palaeontology/evolution/taxonomy and systematics

dentitions, early vertebrates, acanthodians, chondrichthyans, tooth evolution, Palaeozoic

**Author for correspondence:**
Sam Giles
e-mail: s.giles.1@bham.ac.uk

# Diverse stem-chondrichthyan oral structures and evidence for an independently acquired acanthodid dentition

## Richard P. Dearden[1] and Sam Giles[2,3]

[1]CR2P, Centre de Recherche en Paléontologie–Paris, Muséum national d'Histoire naturelle, Sorbonne Université, Centre National de la Recherche Scientifique, CP 38, 57 Rue Cuvier, F75231 Paris Cedex 05, France
[2]School of Geography, Earth and Environmental Sciences, University of Birmingham, Birmingham, B15 2TT, UK
[3]Department of Earth Sciences, Natural History Museum, Cromwell Road, London SW7 5BD, UK

RPD, 0000-0003-3522-7304; SG, 0000-0001-9267-4392

The teeth of sharks famously form a series of transversely organized files with a conveyor-belt replacement that are borne directly on the jaw cartilages, in contrast to the dermal plate-borne dentition of bony fishes that undergoes site-specific replacement. A major obstacle in understanding how this system evolved is the poorly understood relationships of the earliest chondrichthyans and the profusion of morphologically and terminologically diverse bones, cartilages, splints and whorls that they possess. Here, we use tomographic methods to investigate mandibular structures in several early branching 'acanthodian'-grade stem-chondrichthyans. We show that the dentigerous jaw bones of disparate genera of ischnacanthids are united by a common construction, being growing bones with non-shedding dentition. Mandibular splints, which support the ventro-lateral edge of the Meckel's cartilage in some taxa, are formed from dermal bone and may be an acanthodid synapomorphy. We demonstrate that the teeth of *Acanthodopsis* are borne directly on the mandibular cartilage and that this taxon is deeply nested within an edentulous radiation, representing an unexpected independent origin of teeth. Many or even all of the range of unusual oral structures may be apomorphic, but they should nonetheless be considered when building hypotheses of tooth and jaw evolution, both in chondrichthyans and more broadly.

# 1. Introduction

The structure and position of teeth and jaws are among the major anatomical distinctions between crown osteichthyans (bony fishes: ray-finned fishes, lobe-finned fishes and tetrapods) and crown chondrichthyans (cartilaginous fishes: sharks, rays and chimaeras) [1]. In osteichthyans, teeth are partially resorbed at their base, shed and replaced in position on dermal bones lateral to and overlying endoskeletal jaw cartilages as part of outer and inner dental arcades. In crownward chondrichthyans, teeth grow, shed and are replaced in parallel rows of labiolingually directed series directly on the jaw cartilages. The origins of these dental structures can be traced back to Palaeozoic taxa, which suggest that the last common ancestor of jawed fishes (gnathostomes), as well as crownward stem-gnathostomes (a paraphyletic assemblage referred to as 'placoderms'), possessed non-shedding teeth fused to the underlying dermal jaw bone [2–7]. However, oral structures in many Palaeozoic gnathostomes remain poorly characterized, and as a result their relevance to the evolution of teeth is unclear.

The advent of micro-computed tomography has led to a renewed interest in tooth evolution and development in Palaeozoic gnathostomes. These have mostly focused on stem-gnathostome 'placoderms' [2,5,6,8] and osteichthyans [9–14] and have revealed an unexpected range of morphologies. Stem-group gnathostomes have non-shedding dentitions, which may be arranged radially [5,8] or in parallel rows [6], borne on an underlying dermal bone. The homology of the dermal jaw bones in stem-group gnathostomes to the inner and outer dental arcades of crown-gnathostomes is uncertain [6,15,16]. Meanwhile, many Palaeozoic osteichthyans possessed shedding dentitions comparable to more recent taxa, although some stem osteichthyans have dental structures such as symphyseal tooth whorls [14] and marginal cusps organized into rows [9–13], which are more broadly comparable to the gnathostome total group. Early-branching members of the chondrichthyan total group (including 'acanthodians') have received less attention, with only a handful of taxa described using CT data [7,17–20]. This is despite a remarkable array of dermal oral structures across the assemblage: various early chondrichthyans possess tooth whorls [17,21,22], gracile or molariform teeth not organized into whorls [18,23,24], dermal plates of differing constrictions with and without teeth [22,23,25–28], or may lack dermal mandibular structures entirely [29]. A variety of extramandibular 'dentitions' and other oral structures are also known [29,30].

Teeth arranged into files are widespread in chondrichthyans both living and extinct. In the larger of the two constituent chondrichthyan clades (elasmobranchs: sharks and rays), teeth are continuously replaced in generative series [1]. They grow on the inner margin of the jaw, move through a labiolingual file (figure 1a) and are shed at the labial jaw margin [1]. In holocephalans (chimaeras and relatives), the dentition is modified to two upper pairs and one lower pair of non-shedding, hypermineralized toothplates [35]. The elasmobranch-like condition of labiolingual files of teeth is seen in both stem-group elasmobranchs [36] and stem-group holocephalans [37], implying that this condition is plesiomorphic for chondrichthyans. Many Palaeozoic chondrichthyans possessed tooth whorls, where the tooth file comprises multiple cusps fused onto a common base. Tooth whorls are common in taxa in the chondrichthyan stem-group, and may form the entire dentition (e.g. *Ptomacanthus* [21]; *Doliodus* [17]) or be present at and/or restricted to the symphysis (e.g. ischnacanthids: [20,22]). It is unclear whether individual teeth were shed from tooth whorls borne on mandibular rami, or whether the whorls themselves were shed [17]. Some probable crown-group chondrichthyans had a further condition in which teeth did not share a common base but post-functional teeth were retained at the labial margin of the jaw [38]. This, alongside the prevalence of tooth whorls with fused teeth in stem-chondrichthyans, indicates that tooth shedding was acquired later in the chondrichthyan total group.

In addition, a ream of stem-group chondrichthyans have unusual dentitions that do not conform to a file-like arrangement, and which are less commonly considered in hypotheses of tooth evolution (figure 1b–f). The most well-characterized of these are dentigerous jaw bones: large, tooth-bearing dermal plates that sit on both the upper and lower jaw cartilages in 'ischnacanthid' stem-group chondrichthyans [22]. Dentigerous jaw bones bear one or more rows of anteroposteriorly aligned teeth, the cusps of which can be quite morphologically and presumably functionally variable [25]. The tooth rows have been shown to grow via anterior addition of new cusps in one taxonomically unidentified specimen, but construction of the bone beyond the tooth rows, as well as the array of different morphologies, remain poorly characterized [7,26,27]. Other stem-group chondrichthyans, such as *Gladbachus* [24], *Pucapampella* [18] and *Acanthodopsis* [23], have teeth that are neither part of dentigerous jaw bones nor arranged into files. Diplacanthid stem-group chondrichthyans [27,28] bear smooth, toothless dermal plates on their lower jaws. Numerous stem-group chondrichthyans lack

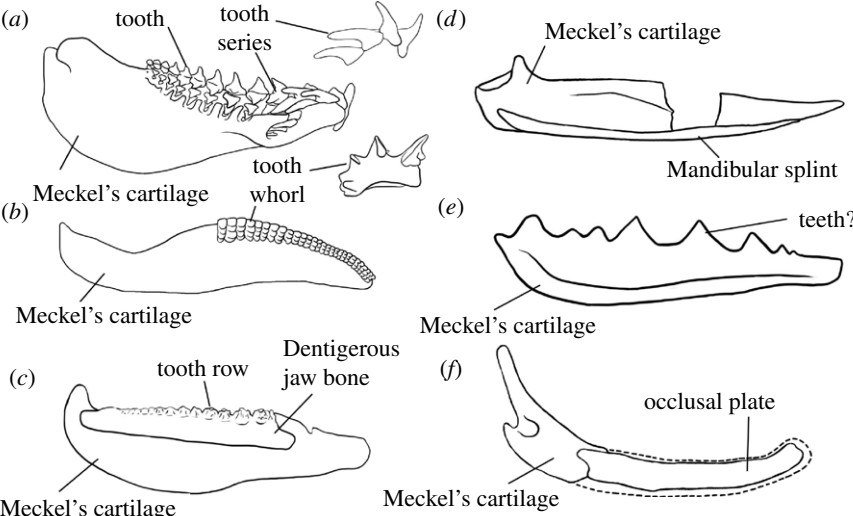

**Figure 1.** Summary figure showing lower jaws of six chondrichthyans, and major structures discussed in the text. (*a*) The extant elasmobranch *Isurus* in medial view and with tooth series in lateral view, drawn from three-dimensional data [31]; (*b*) the stem-chondrichthyan *Ptomacanthus* in medial view reconstructed from [32] and with tooth whorl from [21]; (*c*) the ischnacanthid stem-chondrichthyan *Erymnacanthus* in medial view redrawn from [25]; (*d*) the acanthodid stem-chondrichthyan *Acanthodes* in lateral view redrawn from [33]; (*e*) the acanthodid stem-chondrichthyan *Acanthodopsis* in lateral view redrawn from [23]; (*f*) the diplacanthid stem-chondrichthyan *Diplacanthus* in medial view reconstructed from [61] and [28]. Not to scale.

teeth altogether, and the lower jaws of many of these taxa bear poorly characterized mandibular splints [30,39] which have also been identified in some diplacanthids and ischnacanthids [40,41].

Further confounding this diversity of oral structures are competing and unsettled hypotheses of relationships for early chondrichthyans. While 'acanthodians' are now established as stem-group chondrichthyans [24,32,36,37,42–45], there is limited certainty over the monophyly of 'acanthodian' subgroups. There is some evidence that an assemblage of diplacanthiform, ischnacanthiform and acanthodiform taxa may form a clade or grade subtending the remainder of the chondrichthyan total group [24]. 'Climatiid' acanthodians, which have overlapping character complements with more shark-like chondrichthyans [18], tend to be recovered in a more crownward position (although see [46]). Beyond this, however, different phylogenetic analyses present very different schemes of relationships, often with low support. These conflicting patterns of relationships present major obstacles to understanding patterns evolution for many of the dental structures seen in early chondrichthyans, although likelihood-based methods provide a possible approach [7].

Here, we use computed tomography to image the teeth, jaws and associated oral structures of several early diverging stem-group chondrichthyans with the aim of more broadly sampling the diversity of their oral structures. We aim to characterize the anatomy of several different ischnacanthid dentigerous jaw bones of different constructions, the mandibular splint of acanthodids and the 'teeth' of *Acanthodopsis*. We contextualize our new data within a wider review of stem-group chondrichthyan oral structures and identify those that represent synapomorphies, while also discussing challenges in reconstructing dental evolutionary histories and homologies.

# 2. Material and methods

## 2.1. Taxa examined

All specimens studied here are housed at the Natural History Museum, London (NHMUK), and comprise: an isolated jaw of *Taemasacanthus erroli* (NHMUK PV P.33706); an isolated jaw of *Atopacanthus* sp. (NHMUK PV P.10978); an isolated jaw of *Acanthodopsis* sp. (NHMUK PV P.10383); a partial head of *Acanthodes* sp. (NHMUK PV P.8065) and an isolated jaw of *Ischnacanthus* sp. (NHMUK PV P.40124).

*Taemasacanthus erroli* is known from eight isolated jaws from the Emsian (Lower Devonian) Murrumbidgee Group in New South Wales, Australia [47]. Other species, also based on isolated jaw

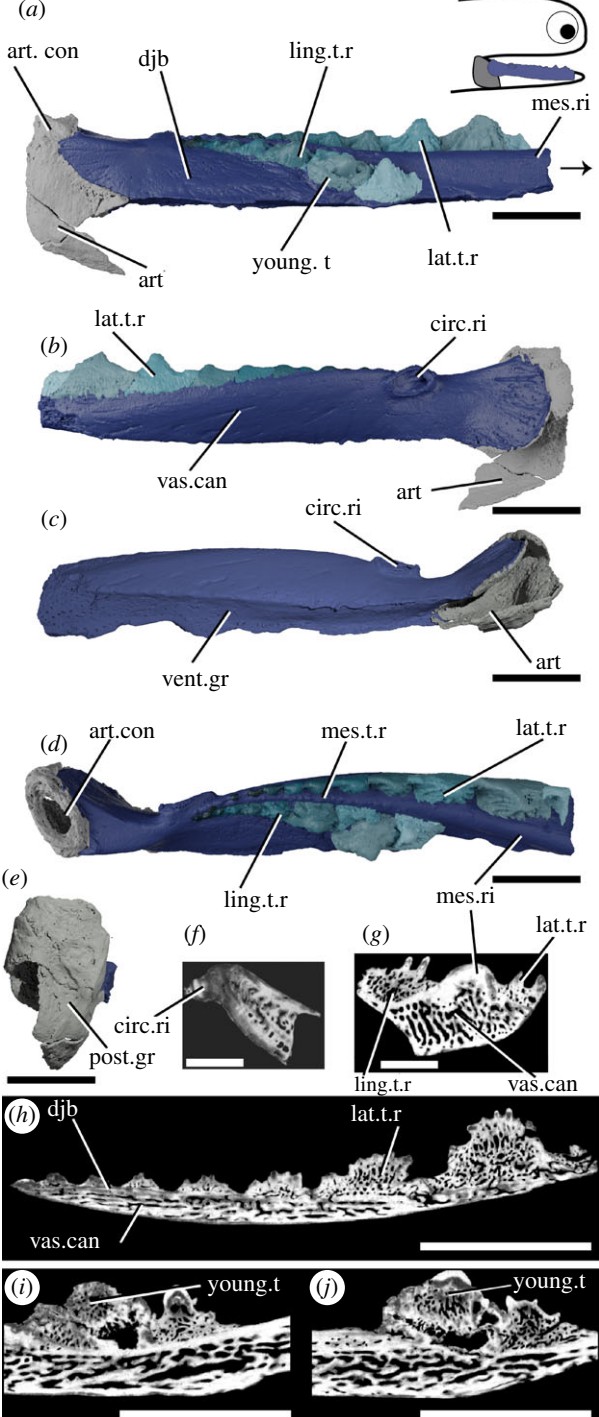

**Figure 2.** Tomographic model of the left lower jaw of *T. erroli* NHMUK PV P.33706 in (*a*) medial view, (*b*) lateral view, (*c*) ventral view, (*d*) dorsal view, (*e*) posterior view, and reconstructed tomograms showing (*f*) a transverse section through the circular ridge, (*g*) a transverse section through the lower jaw, (*h*) a sagittal section through the lingual tooth row and (*i,j*) progressively medial sagittal sections through the aberrant youngest cusp and neighbouring cusps. Teeth in (*a*), (*b*) and (*d*) are coloured separately from the dentigerous jaw bone. Arrow indicates direction of anterior, and top right inset shows location of rendered jaw components. art, articular (Meckel's cartilage); art.con, articular 'condyle'; circ.ri, circular ridge; djb, dentigerous jaw bone; lat.t.r., lateral tooth row; ling.t.r, lingual tooth row; mes.ri, mesial ridge; mes.t.r, mesial tooth row; post.gr, posterior groove; young.t, out-of-order youngest tooth; vent.gr, ventral groove; vas.can, vascular canals. Scale bar = 5 mm.

bones, have been assigned to the genus [48,49], but no articulated animals are known. *Taemasacanthus* is understood to be an ischnacanthid on the basis of its dentigerous jaw bone [47]. NHMUK PV P.33706, described here (figure 2), is a right lower jaw, and comprises two main parts: a dermal dentigerous jaw bone and the articular ossification of Meckel's cartilage. The external morphology of this

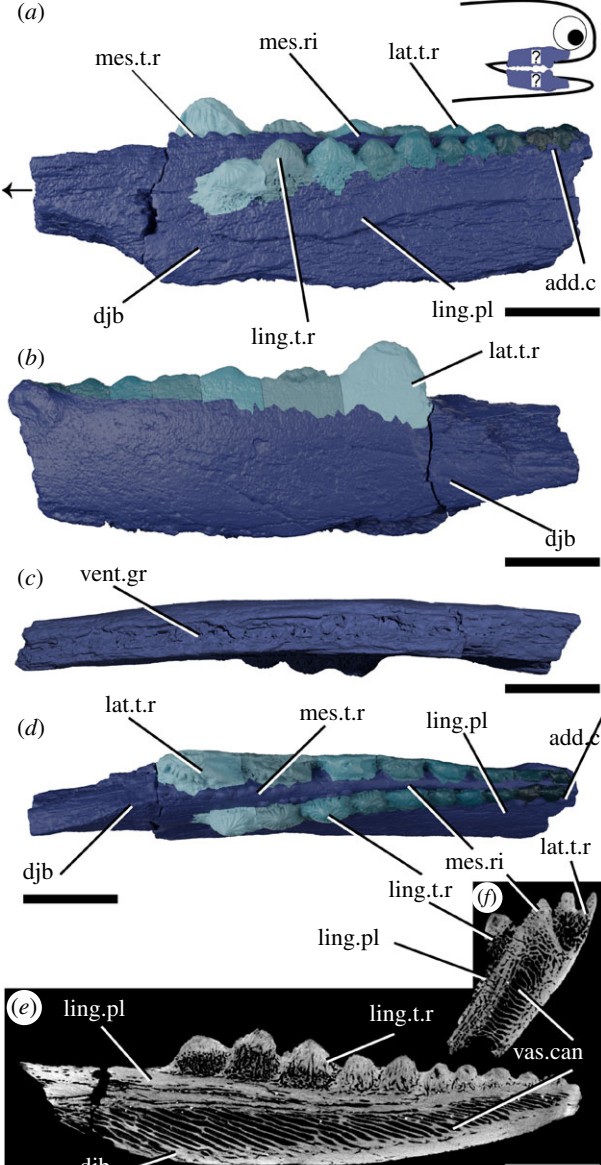

**Figure 3.** Tomographic model of a dentigerous jaw bone of *Atopacanthus* sp. NHMUK PV P.10978 in (*a*) medial view, (*b*) lateral view, (*c*) ventral view, (*d*) dorsal view and a reconstructed tomograms showing (*e*) a sagittal section through the lingual tooth row and (*f*) a transverse section through the lower jaw. Arrow indicates direction of anterior, and top right inset shows possible locations of rendered jaw components. Teeth in (*a*), (*b*) and (*d*) are coloured separately from the dentigerous jaw bone. add.c, additional cusps; djb, dentigerous jaw bone; lat.t.r., lateral tooth row; ling.pl, lingual plate; ling.t.r, lingual tooth row; mes.ri, mesial ridge; mes.t.r, mesial tooth row; vent.gr, ventral groove; vas.can, vascular canals. Scale bar = 5 mm in (*a*–*e*) and (*h*–*j*), 2 mm in (*f*–*h*).

specimen of *Taemasacanthus* has been fully described [47] but is briefly redescribed here to contextualize our new information. An additional ventral fragment of the articular (previously figured [47]) has become detached from the rest of the ossification and was not included in the CT scan.

Atopacanthus is known throughout the Middle–Upper Devonian [50,51]. The type species, *Atopacanthus dentatus*, is known from several dentigerous jaw bones from near Hamburg, New York and is presumed to be an ischnacanthid [23,52]. The sole articulated specimen attributed to *Atopacanthus* sp., from the Upper Devonian of the Rhineland [50,53], has since been referred to *Serradentus* [54] and so the genus is known only from disarticulated remains. The specimen described here, NHMUK PV P.10978 (figure 3), is a dentigerous jaw bone collected from Elgin, Scotland and is Late Devonian in age. It was originally labelled as a possible dipnoan toothplate before later being referred to *Atopacanthus*, and its morphology conforms with that of other specimens described as *Atopacanthus*. It is not associated with any endoskeletal material, and it is impossible to tell whether it is from a right lower jaw or left upper jaw. For ease of comparison with other specimens, we describe its morphology as if it were a part of the lower jaw.

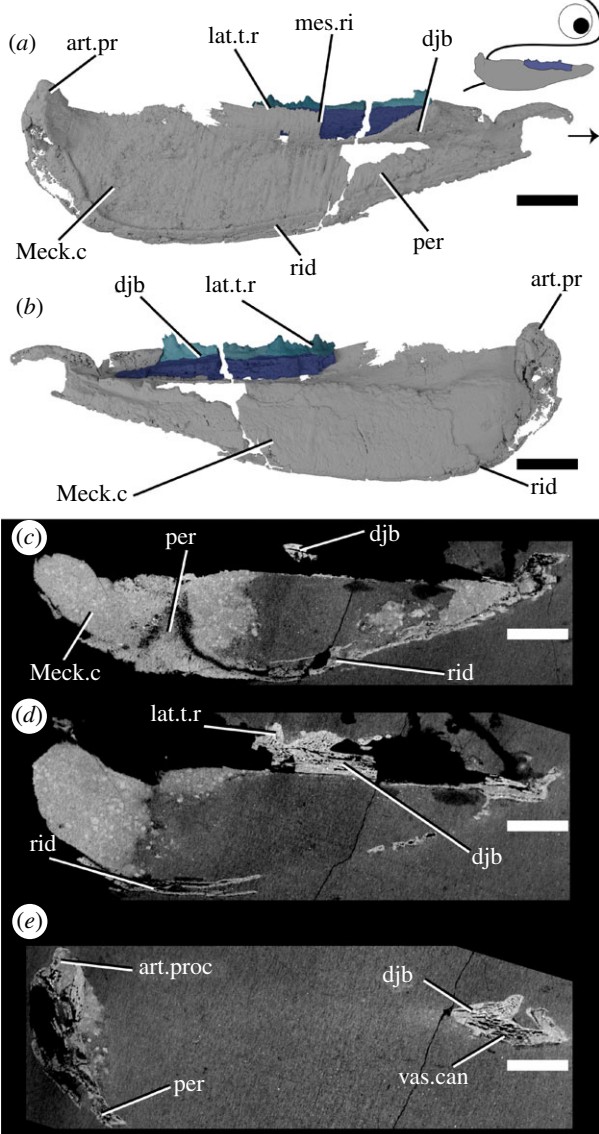

**Figure 4.** Tomographic model of the left lower jaw of *Ischnacanthus* sp. NHMUK PV P.40124 in (*a*) lateral view, (*b*) medial view, and (*c*), (*d*), (*e*) reconstructed tomograms showing successively deeper sagittal sections. Arrow indicates direction of anterior, and top right inset shows location of rendered jaw components. art.proc, articular process; djb, dentigerous jaw bone; lat.t.r., lateral tooth row; Meck.c, Meckel's cartilage; mes.ri, mesial ridge; per, perichondral bone; rid, ridge; vas.can, vascular canal. Scale bar = 5 mm.

*Ischnacanthus* is the best known ischnacanthid 'acanthodian', represented by numerous articulated specimens of Lochkovian (Lower Devonian) age from the Midland Valley in Scotland [22]. The material described here (NHMUK PV P.40124; figure 4) is an isolated left lower jaw from the Lochkovian Midland Valley in Tealing, Forfarshire [55]. It is fairly complete, but parts of the dorsal and anterior margins have been lost to the counterpart (which is preserved, but was not CT scanned), and the whole jaw is laterally flattened. It comprises a dentigerous jaw bone and Meckel's cartilage.

*Acanthodopsis* is known from the Carboniferous of the UK and Australia. *Acanthodopsis* has been previously considered an ischnacanthid on the basis of its 'dentigerous jaw bones', but in terms of its skeletal anatomy it is more similar to acanthodids [23,47]. The material described here, NHMUK PV P.10383 from the Northumberland Coal Measures (figure 5), comprises a laterally flattened lower right jaw, consisting of a Meckel's cartilage with teeth and a mandibular splint.

*Acanthodes* is the latest occurring genus of 'acanthodian' found as articulated body fossils from the Mississippian (Carboniferous) into the Lower Permian [35]. It is the only genus of 'acanthodian'-grade animal known from extensively preserved endoskeleton, seen in specimens of *Acanthodes confusus* from Lebach, Germany [33]. The material described here (NHMUK PV P.8065) comprises part of the ventral half of the head of a specimen from the Knowles Ironstone of Staffordshire (figure 6). As the

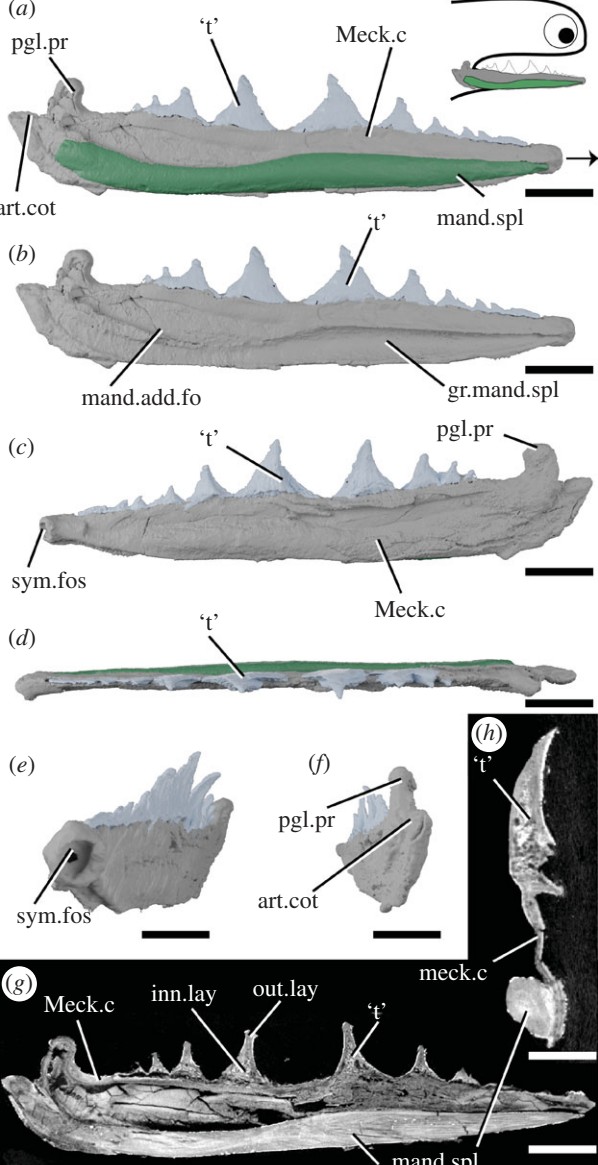

**Figure 5.** Tomographic model of the right lower jaw of *Acanthodopsis* sp. NHMUK PV P.10383 in lateral view with (*a*) and without (*b*) mandibular splint, (*c*) medial view, (*d*) dorsal view, (*e*) anteromedial view, (*f*) posterior view, and reconstructed tomograms showing (*g*) a sagittal section through the entire jaw and (*h*) a transverse section through the jaw. Arrow indicates direction of anterior, and top right inset shows location of rendered jaw components. art.cot, articular cotylus; gr.mand.spl, groove for mandibular splint; inn.lay, inner layer; mand.add.fo, mandibular adductor fossa; mand.spl, mandibular splint; Meck.c, Meckel's cartilage; out.lay, outer layer; pgl.pr, preglenoid process; sym.fos, symphyseal fossa; 't', 'teeth'. Scale bar = 5 mm in (*a–g*), 2 mm in (*h*).

dorsal margins of the jaw bones are obscured within the rock, it was originally referred to '*Acanthodopsis* or *Acanthodes*'. As CT scanning shows that dentition is absent, we can confirm it to be *Acanthodes* sp. Most of the left jaw is preserved, and of the right jaw only the mandibular splint is preserved, as are some of the lower branchiostegal ray series and isolated dermal gill rakers. Scattered parts of the rest of the head endoskeleton are also present, including parts of the ceratobranchials and a hyomandibular.

## 2.2. CT scanning

Full details of scanning parameters are given in electronic supplementary material, table S1. The voxel sizes for each scan are as follows: *Acanthodes*, 44.9 µm; *Acanthodopsis*, 22.6 µm; *Atopacanthus*, 19.51 µm; *Ischnacanthus*, 24.6 µm; *T. erroli*, 17.3 µm.

Reconstructed tomographic datasets were segmented in Mimics v. 19 (biomedical.materialise.com/mimics; Materialise, Leuven, Belgium) and images were generated using Blender (blender.org).

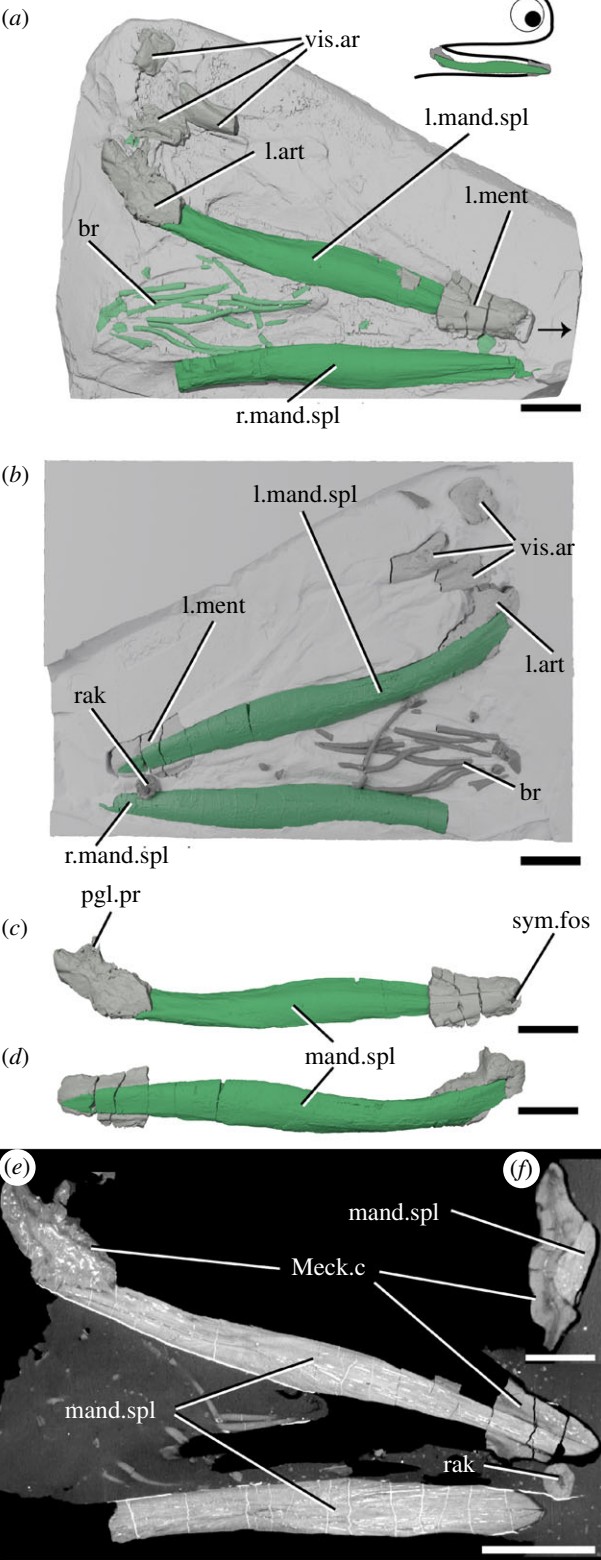

**Figure 6.** The lower jaws of *Acanthodes* sp. NHMUK PV P.8065 in (*a*) dorsal view against the matrix, (*b*) in ventral view superimposed on a digital mould of the matrix's surface, the left lower jaw isolated in (*c*) medial and (*d*) lateral view, and reconstructed tomograms showing (*e*) a coronal section through the specimen and (*f*) a transverse section through a lower jaw. Arrow indicates direction of anterior, and top right inset shows location of rendered jaw components. art, articular (Meckel's cartilage); br, branchiostegal rays; mand.spl, mandibular splint; Meck.c, Meckel's cartilage; ment, mentomandibular (Meckel's cartilage); pgl.pr, preglenoid process; rak, gill raker; sym.fos, symphyseal fossa; vis.ar, visceral arch fragments. Scale bar = 5 mm in (*a–e*), 2 mm in (*f*).

## 2.3. Phylogenetic analyses

Our dataset is based on Dearden *et al*. [32], the most recently published phylogenetic dataset specifically tackling stem-group chondrichthyan relationships. We have added one taxon and four characters and made minor modifications to some codes (full details given in the electronic supplementary material). We performed a parsimony analysis in TNT [56] with the following settings: a parsimony ratchet with 10 000 iterations, holding 100 trees per iteration, with TBR branch swapping. Galeaspida was set as the outgroup and the following constraint applied: (Galeaspida(Osteostraci(Mandibulate Gnathostomes))). We also performed a Bayesian analysis, with Galeaspida set as the outgroup, mandibulate gnathostomes constrained as monophyletic and the following settings. We used a uniform prior with the Mkv model and gamma-distributed rates; searched for 10 000 000 generations, sampling every 1000 generations; and calculated the majority-rule consensus tree with a relative burn-in of 25%.

# 3. Results

## 3.1. Taemasacanthus

The dentigerous jaw bone of *Taemasacanthus* is approximately half the full depth of the jaw and is sinusoidal in dorsal view (figure 2). A circular ridge, previously suggested to be for a labial cartilage attachment [47] but more likely the attachment site for a ligament, is present on the lateral surface approximately 1/4 of the way along its length (figure 2*b,c*). Posteriorly, the bone curves laterally and broadens to wrap around the articular. The lateral expansion is larger than the medial expansion, and both are rounded posteriorly. Ventrally, a groove formed by the posterior confluence of these two processes runs underneath the entire length of the dentigerous jaw bone and would have overlain Meckel's cartilage (figure 2*c*). The dentigerous jaw bone is approximately trapezoid in cross-section and bears three rows of teeth (figure 2*a,b,d,g,h*). Histologically the underlying dermal plate comprises heavily vascularized dermal bone, similar to that observed in thin sections of other ischnacanthid dentigerous jaw bones [7,22,57], with a relatively thin layer of less vascularized bone around the plate margins (figure 2*g*; electronic supplementary material, figure S1a–d). The vascularization comprises an interlinked network of tubules, which are strongly anteroposteriorly polarized in the tooth-bearing section of the bone. Vascular channels occasionally open onto the surface of the bone, particularly the ventral groove. On the lingual surface of the bone, a change in orientation of the vascularization (from anteroposterior to more random) indicates the presence of a separate plate-like unit of growth (electronic supplementary material, figure S1d). The more posterior wrapped part of the bone has a radial arrangement of vasculature suggesting that the bone grew posteriorly and ventrally as well as anteriorly as the underlying endoskeletal jaw grew. The circular ridge is formed of avascular bone, but is otherwise a similar tissue to that forming the outer margin of the dermal plate (figure 2*f*).

Three rows of teeth are borne on the biting edge of the dermal plate, all starting approximately at the level of the mesial ridge: a lateral, medial and lingual row (figure 2*a,b,d*). Teeth of the medial row are far smaller and less distinct than those of the lateral and lingual rows. Teeth within the lateral and lingual rows are fused to the jaw, but the base of the tooth is marked by an increase in the density of random and dorsally oriented vascular canals (figure 2*f*; electronic supplementary material, figure S1a–c). The medial row lies on the mesial ridge and comprises a single row of small disorganized cusps that are continuous with the underlying dermal plate and do not appear to overlap with each other. Vasculature in the medial row is only visible in the more posterior cusps and does not seem related to size (electronic supplementary material, figure S1f). The lateral and lingual rows of teeth are much larger and ridged, with a vascular base topped with a mostly avascular crown. The younger, larger, more anterior teeth of both rows have extensively vascularized crowns. The smaller, more posterior cusps have less—or even no—vascularization, indicating that the crown was infilled in older teeth. The basal vasculature has occasional connections with the vasculature of the underlying dermal plate. The teeth seemingly lack a continuous enameloid covering (figure 2*f*), contra recent reports of enameloid in an ischnacanthid [7], although we caution that this may be due to the resolution of our dataset. Both tooth rows grow by the addition of new teeth onto the anterior end of the row, as evidenced by anterior teeth partially overlying posterior ones, and cusps becoming progressively larger in the direction of growth. The sole exception to this is in the lingual tooth row, where, in what is probably a pathology, the eighth cusp of the lingual row is incomplete and is either damaged or its growth has been aborted (figure 2*a*). Both the damaged eighth cusp and the undamaged tenth cusp are

overgrown by the youngest cusp in the row. This, the ninth cusp, is oriented notably more medially than other cusps and may have disrupted the growth of the smaller tenth cusp, although the underlying bone appears unaffected. The vasculature of the ninth cusp appears to be isolated from the surrounding vasculature, while the vasculature of the underlying eighth cusp opens into a large, central hollow. The lateral tooth row comprises around 12 cusps, and its teeth are laterally unornamented and continuous with the lateral surface of the dermal bone, connected to one another via anteroposterior lateral ridges (figure 2b). The lingual side of each cusp is rounded and ornamented with a number of ridges, which become longer and progressively more tuberculated on more anterior cusps. The lingual tooth row comprises 10 cusps, which curve away from the occlusal surface anteriorly.

Only the posteriormost portion of the Meckel's cartilage, the articular ossification, is preserved (figure 2). It is formed from a sheath of perichondral bone and would have been filled with cartilage in life. Some spongy texture is apparent on the interior surface of the perichondral bone. A shallow groove on the posterior surface does not appear to continue ventrally, making it unlikely to have accommodated a mandibular splint as previously suggested [47]. Articulation with the palatoquadrate appears to be via an open, oval, fossa [47] (figure 2d). The tissue forming this is notably ill-formed, and appears to lack a solid perichondral covering; as Burrow et al. [22] suggest it seems likely to be an articular process like that in other ischnacanthids [22,25] which is broken.

## 3.2. Atopacanthus

The dentigerous jaw bone is robust, trapezoid in cross-section, but flattened laterally and taller proportionate to its length compared to Taemasacanthus (figure 3). It is slightly medially convex. The anterior fifth of the preserved bone (it is broken both anteriorly and posteriorly) is toothless and tapers slightly. A narrow, shallow groove to accommodate the mandibular cartilage runs along its ventral surface (figure 3c). The histology is similar to Taemasacanthus, with heavily vascularized dermal bone surrounded by a less vascular layer, but the vascular tubules are even more strongly polarized in an antero-dorsal direction (figure 3e; electronic supplementary material, figure S1e–g). Again, the vasculature principally opens into the ventral groove. Towards the surface, bone vascularization is less dense, and not polarized. The lingual face of the dentigerous jaw bone supports a distinct thin, lingual, tooth-bearing plate (figure 3a,d,e), which is comparable in position and vascularization to the plate-like region in Taemasacanthus (electronic supplementary material, figure S1d). This plate is still heavily vascularized, but tubes are polarized dorso-lingually. On the outer perimeter of the lateral face of the main bone, the vasculature is oriented obliquely.

As in Taemasacanthus, lateral, mesial and lingual tooth rows are borne on the dorsal surface of the underlying dermal plate (figure 3a,b,d). The medial ridge bears two disorganized rows of cusps along its anterior half, with the posterior half being smooth. All cusps are vascular and are histologically continuous with the medial ridge. The lateral tooth row comprises eight cusps, which become progressively larger anteriorly, and their lateral surfaces are continuous with the outside of the dermal plate (figure 3b). Their lingual surfaces are rounded and ornamented with untuberculated ridges. The lingual tooth row comprises 10 main cusps, which curve medially across the dentigerous jaw bone. Two additional small cusps are present near the posterior margin of the dermal plate, ventral to the main lingual tooth row, which are closest in appearance to the teeth of the medial ridge (figure 3a,d). The lingual tooth row lies on top of a lingual plate, which is apposited onto the lingual surface of the main dermal plate (figure 3a,d,e). As in the lateral row and in Taemasacanthus, cusps become larger anteriorly and are ornamented with ridges. The histology of the teeth of the lateral and lingual rows comprises a vascular base topped with an avascular cap lacking enameloid, with the vascular canals oriented distinctly from the underlying dermal and lingual plates (figure 3e). As in Taemasacanthus, younger teeth are more heavily vascularized with an extensively vascularized crown, and the anteriormost tooth still posseses a clear pulp cavity (electronic supplementary material, figure S1e–g), suggesting that tooth vasculature became infilled with age. Teeth in both the lateral and lingual rows were added anteriorly, with anterior cusps partly overlying their posterior fellows.

## 3.3. Ischnacanthus

Only the anterior part of the dentigerous jaw bone is preserved in the part (figure 4), although the mould of the posterior region is visible in outline. The underlying dermal plate is much shallower than in Taemasacanthus and Atopacanthus. A lateral tooth row and a medial ridge are present. The lateral tooth row preserves four cusps, the third of which is exposed on the surface and, therefore, incomplete dorsally. The cusps are linked by a cuspidate ridge along their lateral faces. Relative size and age are

difficult to determine due to the mode of preservation, but the anteriormost cusp is the largest, and cusp overlap indicates that teeth were added anteriorly. Although the ventralmost parts of the dermal plate are missing, the ventral margin of the teeth is marked by a notable shift in density and orientation of the vascular canals (figure 4d). The tissue forming the teeth is similar to *Taemasacanthus* and *Atopacanthus*, with a vascularized base and an avascular crown apparently lacking enameloid [22]. The internal vasculature of the bone in *Ischnacanthus* is also longitudinally polarized and connected, although less well-visualized in our scan data (electronic supplementary material, figure S1 h,i).

The large Meckel's cartilage is near-complete and preserved as a single ossification (figure 4). It is curved posteriorly and tapers anteriorly. The dentigerous jaw bone is borne on its dorso-lingual surface. A laterally directed articular condyle is present at the posterior extent, and a shallow groove extends ventral to the condyle. The majority of Meckel's cartilage is formed of globular calcified cartilage. Parts of its lateral surface, as well as its ventral, anterior and posterior extents, are covered by a thin, densely mineralized tissue that appears to be perichondral bone [22]. This tissue thickens ventrally and posteriorly and is fractured. The perichondral sheath is avascular but has fractured in such a way that cracks and voids artificially resemble the vasculature of the dentigerous jaw bone. A thickened ridge along the posteroventral and posterior margin is continuous with the perichondral rind that extends onto the lateral surface, but externally gives the appearance of a separate ossification (figure 4b,c). In section, this is closely comparable to the so-called 'mandibular bone' that Ørvig [41] described in *Xylacanthus* and is probably responsible for accounts of mandibular splints in ischnacanthids.

## 3.4. Acanthodopsis

The lower jaw in *Acanthodopsis* comprises a tooth-bearing Meckel's cartilage and a mandibular splint (figure 5). The Meckel's cartilage is long and thin and similar in form to that of *Acanthodes* (figure 5; [23,33]), with an identical articular cotylus and marked preglenoid process. It tapers anteriorly, terminating in a small, cup-shaped anterior symphyseal fossa (figure 5c,e). The Meckel's cartilage is formed from a shell of what we infer to be perichondral bone [23], which has collapsed and cracked under compression and appears to be unmineralized internally (figure 5g,h), although some mineralization appears to be present in the jaw articulation. Unlike *A. confusus*, it is perichondrally mineralized along its entire length (figure 5), rather than in separate articular and mentomandibular sections.

Ten monocuspid, triangular teeth form a row along the dorsal surface of the Meckel's cartilage (figure 5). The largest tooth is in the middle of the jaw, with teeth becoming smaller and more closely set anteriorly and posteriorly; they are slightly lingually convex, each with a smooth (but possibly weathered) lateral face and a longitudinally striated lingual face. Previous descriptions of the teeth [23] were undecided as to their tissue makeup. The tissue comprising them comprises distinct inner and outer layers. The outer layer is thick and covers the outside surface of each tooth but does not close ventrally: unlike the perichondral surface of the cartilage it is not crushed and the surface appears intact. The inner layer has a spongy texture, which may reflect internal vasculature: no obvious pulp canals are present. Although the contrast between the teeth and jaw bone is subtle, they can be differentiated in that the internal tissues of the teeth have a spongy texture, whereas the perichondral bone is solid. We infer the teeth to be dermal due to this histological distinction from the Meckel's cartilage as well as their gross structure, which shows a separation from the Meckel's cartilage, and their ornamentation. We consider it likely that these tissues are dentinous but more detailed study is needed to establish their identity. The direction of growth is difficult to infer. In successive tomograms viewed in sequence, the largest tooth appears to be overlapped by the anterior and posterior teeth, possibly making it the oldest. This contrasts to the order of growth in *Taemasacanthus*, *Atopacanthus* and *Ischnacanthus*, where teeth are added anteriorly. However, as with the identity of the tissue, we express caution at this interpretation.

The mandibular splint in *Acanthodopsis* is an unornamented, slightly sinusoidal bone that fits into a groove on the ventro-lateral part of Meckel's cartilage, extending almost its entire length (figure 5a,b). This groove was likely originally much shallower, and its depth has been exaggerated by lateral flattening of the specimen. The tissue forming the splint is solid and is organized into multiple concentric lamellae likely representing lines of arrested growth. It is pierced by a series of thin, longitudinally oriented canals (figure 5g,h). This tissue is distinct from that forming Meckel's cartilage, in particular being denser and better organized, and can be interpreted as dermal bone, especially given the ornamentation of the mandibular splints of some acanthodids [58]. Burrow [23] reported that the mandibular splint in *Acanthodopsis* was formed from cartilage and bone based on thin sections, but the cartilage identified in these reports may have been the Meckel's cartilage above the bone instead (CJ Burrow July 2020, personal communication to R.P.D.).

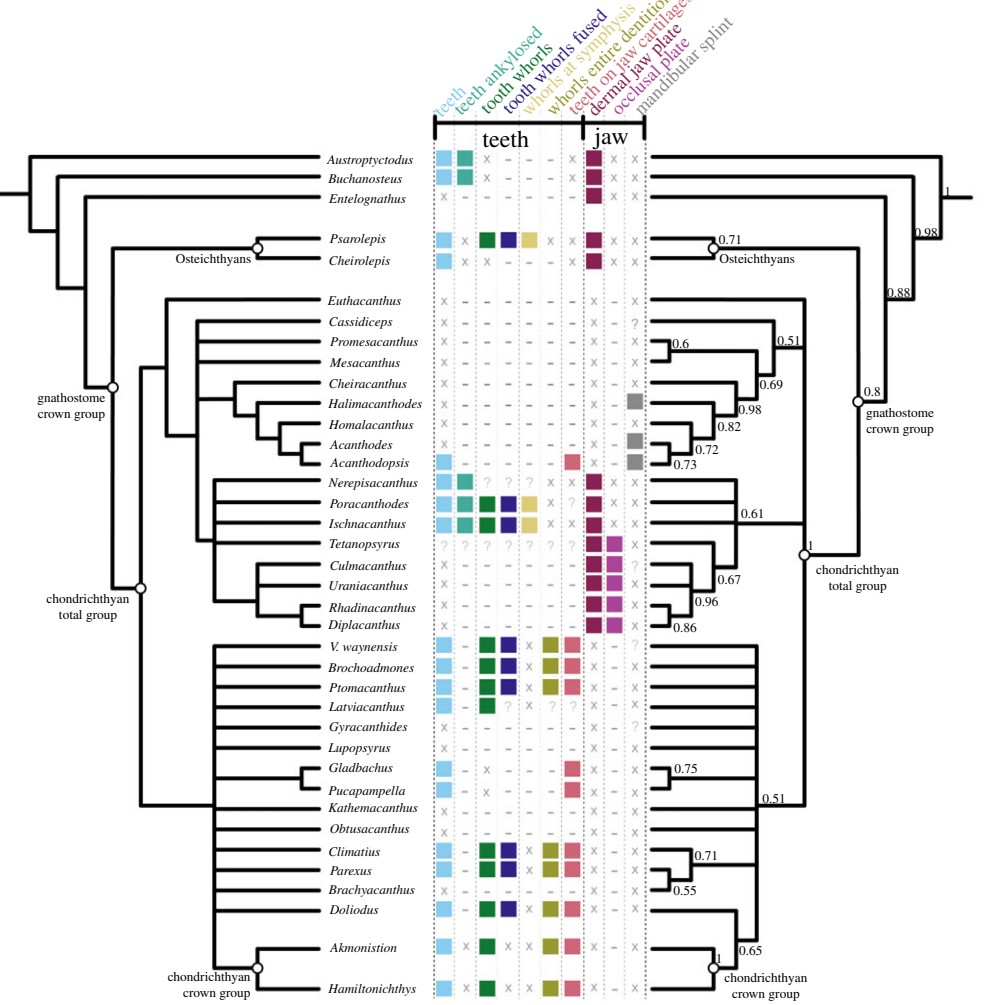

**Figure 7.** Phylogenetic relationships of early chondrichthyans and distribution of oral structures. Strict consensus of 26,101 most parsimonious trees on left and Bayesian analysis on right, with some non-chondrichthyan taxa excluded (full tree with support values in electronic supplementary material, figures S2 and S3). Numbers at nodes on right represent Bayesian posterior probabilities. Character distribtion is based on coding in the data matrix as follows: teeth, character (c.) 82; teeth ankylosed, c.93; tooth whorls fused, c.85; whorls at symphysis and whorls entire dentition, c.88; teeth on jaw cartilages and dermal jaw plate, c.94; occlusal plate, c.269; mandibular splint, c.268. Filled boxes indicate presence of feature; 'x' indicates feature absent; '?' indicates uncertainty; '—' indicates inapplicability.

## 3.5. Acanthodes

The left lower jaw comprises ossified articular and mentomandibular parts of Meckel's cartilage, as well as a mandibular splint (figure 6). Mineralized parts of the Meckel's cartilage are formed from thick perichondral bone and are slightly laterally crushed (figure 6a–d). The articular is as previously described [33]. The mentomandibular has a distinct cup-like symphyseal fossa at its anterior tip, forming part of the mandibular symphysis (figure 6c). The mandibular splint is unornamented, slightly sinusoidal in shape and ellipsoid in cross section. It sits in a groove in the lateral faces of the mentomandibular and articular. Internally, it is solid and vascularized by sparse long, thin canals running its length (figure 6e,f), as in *Acanthodopsis*. A single tooth-like cusp sitting in (although separate from) a cushion-shaped base is probably a branchial or hyoid raker (figure 6b,e).

## 3.6. Phylogenetic results

Our parsimony analysis recovered 26,101 most parsimonious trees with a length of 704 steps. The strict consensus of these results (figure 7a; electronic supplementary material, figure S2) is consistent with other recent analyses in finding all 'acanthodians' to be stem-group chondrichthyans. We recover 'Acanthodii' *sensu* Coates *et al.* [24] as a clade subtending the remainder of the chondrichthyan total

group, with *Euthacanthus* as the sister group to the 'Acanthodii'. However, acanthodiforms (i.e. cheiracanthids, mesacanthids and acanthodids) are paraphyletic. Ischnacanthids plus diplacanthids form a clade, but ischnacanthids themselves are paraphyletic, and diplacanthids are a clade to the exclusion of *Tetanopsyrus*. Remaining stem-chondrichthyan taxa, including climatiids, *Gladbachus* and *Doliodus*, are recovered in a polytomy along with a monophyletic chondrichthyan crown group. Support values aside from the chondrichthyan total group and crown-group nodes are typically low. The Bayesian majority-rule consensus tree (figure 7b; electronic supplementary material, figure S3) is broadly consistent with the parsimony strict consensus tree, although 'Acanthodii' is instead recovered as a polytomy subtending all more crownwards chondrichthyans. *Acanthodopsis* is recovered as the sister taxon to *Acanthodes* in both analyses.

# 4. Discussion

## 4.1. Stem-chondrichthyan oral structures

Our new data show that the dentigerous jaw bones of all ischnacanthids, including articulated taxa such as *Ischnacanthus* and those only known from isolated jaws (e.g. atopacanthids and taemasacanthids), were united by a common construction. These follow the model of tooth growth hypothesized by Ørvig [26] and demonstrated by Rucklin *et al.* [7] on the basis of directional wear and overlapping cusps. These teeth were fused to, but distinct from, the underlying bone, which grew with the endoskeletal component of the jaw. Based on the vasculature of *Taemasacanthus* the dermal bone grew radially from a point posterior to the tooth row. This condition and the positions of tooth rows relative to the underlying dermal plate is common to the three different morphologies of dentigerous jaw bone that we describe and we infer it to have been a common feature of ischnacanthid dentigerous jaw bones The presence of an out of sequence tooth in *Taemasacanthus*, where the youngest tooth has partially overgrown a cusp anteriorly (figure 2), suggests that non-sequential growth was possible in the otherwise ordered tooth rows, likely in response to pathology. Dentigerous jaw bones are broadly comparable with the condition in stem-gnathostomes in the sense that non-shedding teeth are ankylosed to and growing on a basal bone. However, phylogenetic topologies supporting homology between these conditions are in limited supply (figures 7 and 8), and are not upheld in our topology or by more detailed analysis [7].

Although dentigerous jaw bones are typically contrasted with tooth whorls [7], we suggest that dentigerous jaw bones can be usefully interpreted by comparison to anteroposteriorly 'stretched out' tooth whorls. Teeth and tooth-like structures are added directionally across the gnathostome total group and this may be a plesiomorphic feature of gnathostome dentitions [6]. However, tooth rows on dentigerous jaw bones are more comparable to whorls than to these other structures in that tooth files are located in a specific position on an underlying dermal plate, growing in a single direction. This stands in contrast to single-directional, but haphazardly arranged, tooth files reported in stem-gnathostomes and early osteichthyan marginal jaw bones [6,9]. Notably, symphyseal tooth whorls and whorl-like cheek scales in some ischnacanthids could suggest common patterning mechanisms affecting dermal structures in and around the mouth of early chondrichthyans [36]. This organization of tooth files could be apomorphic for chondrichthyans if osteichthyan tooth whorls are optimized as homoplasious [7], and could potentially be a character uniting chondrichthyan dentitions more inclusively than the presence of tooth whorls.

The row of monocuspid 'teeth' borne directly on the Meckelian element of *Acanthodopsis* is unlike that of any other known chondrichthyan or gnathostome. Furthermore, the presence of teeth in a Carboniferous taxon deeply nested within an edentulous radiation (figure 7), the oldest of which are Early Devonian in age, strongly suggests that this represents an independent acquisition of dentition. Previous studies of the jaw of *Acanthodopsis* have interpreted it either as a dentigerous jaw bone [26] or as a perichondrally ossified Meckelian bone with 'teeth' [23]. Our data confirm the latter view and show that the teeth in *Acanthodopsis* are histologically distinct from the underlying perichondral bone and so are presumably dermal ossifications attached to its surface. Our phylogenetic analysis supports the view of Long [47] and Burrow [23] that *Acanthodopsis* is closely related to *Acanthodes*, with the presence of teeth representing the only difference between the genera. A possible morphological comparison to these teeth lies in the branchial and hyoid rakers found in acanthodiform fishes like *Acanthodes*, *Cheiracanthus* and *Homalacanthus* [35,59]. As with the teeth of *Acanthodopsis*, these rakers are conical, sometimes striated [60, fig. 3], and decrease in size from the centre outwards (R. P. Dearden 2021, personal

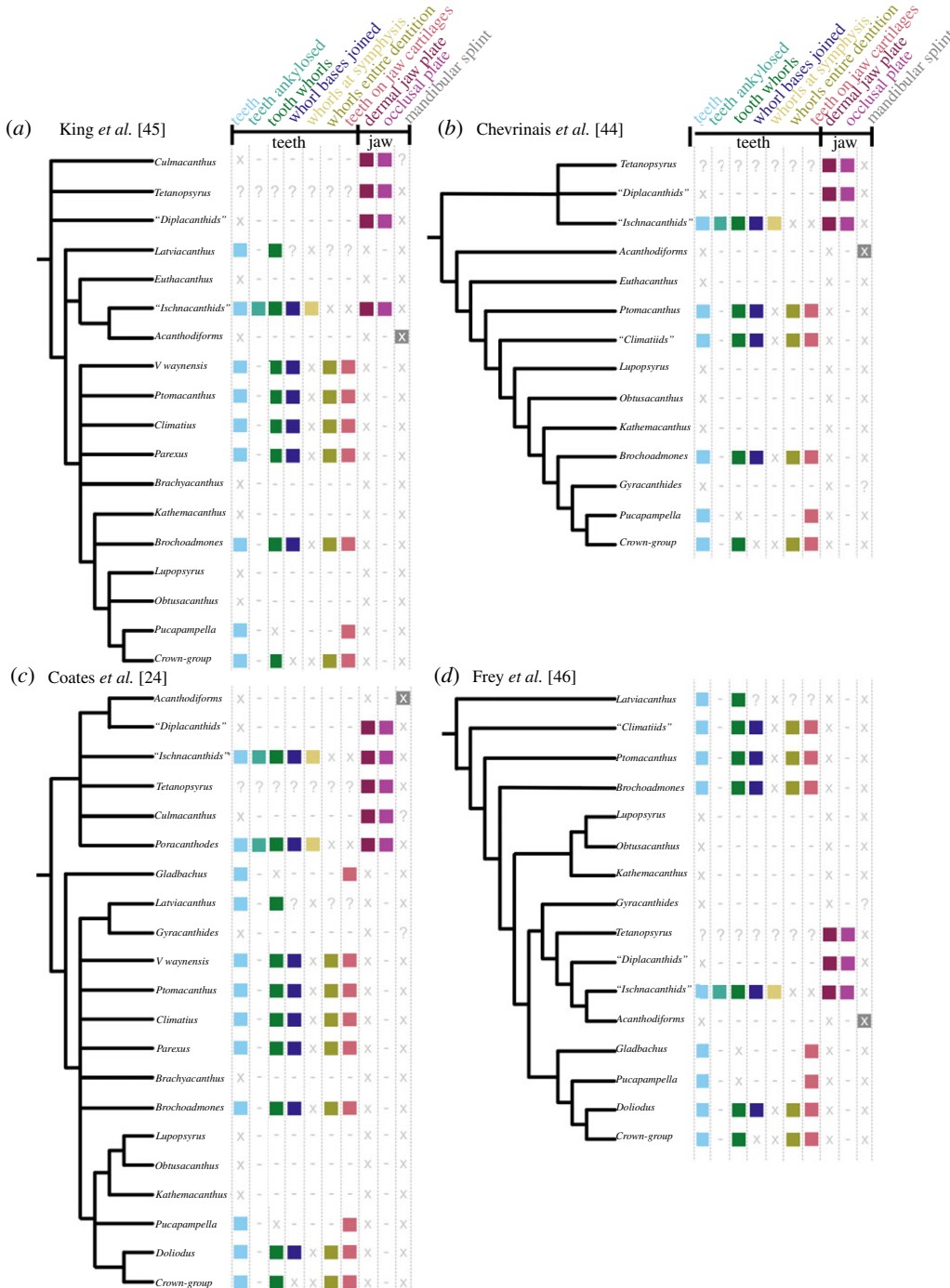

**Figure 8.** Summaries of four contrasting recent phylogenetic schemes of early chondrichthyans, with distribution of oral characters. (a) King et al. [45], (b) Chevrinais et al. [44], (c) Coates et al. [24], (d) Frey et al. [46]. Icons and character numbers as in figure 7. Acanthodiforms includes taxa both with and without mandibular splints.

observations). An alternative to homology between the mandibular dentition in *Acanthodopsis* and other gnathostomes, which is difficult to reconcile with their phylogenetic relationships, may be the co-option of hyoid and branchial rakers to form a novel 'dentition' on the mandibular arch.

## 4.2. Distribution of oral structures in the chondrichthyan stem-group

Teeth and other oral structures in stem-chondrichthyans take on a broad variety of forms, raising questions about their homology and the primitive or derived condition for different features. Here, we briefly review this diversity and map key characters onto our phylogeny.

A diverse array of *teeth* (light blue icons, figures 7 and 8) are present in the majority of Palaeozoic chondrichthyans, including as tooth whorls, but are also remarkable for the breadth of taxa in which they are absent. Acanthodids (e.g. *Acanthodes*: figure 6, [33]) are completely toothless, except for *Acanthodopsis*, as are the likely related mesacanthids (e.g. *Promesacanthus* [39]) and cheiracanthids (e.g. *Cheiracanthus* [59]). In this latter group, tooth-like hyoid rakers have sometimes been mistaken for teeth [61]. Teeth are also absent in diplacanthids [28], with the possible exception of *Tetanopsyrus* (see 'Occlusal plates' section below) [29]. A number of toothless taxa with otherwise diverse anatomies, including *Obtusacanthus*, *Lupopsyrus*, *Euthacanthus*, *Brachyacanthus* and *Kathemacanthus* [34,62–64], are often resolved as more closely related to the crown group (figures 7 and 8). Given the small size and two-dimensional preservation of some of these taxa, it is possible that teeth are present but reduced and so far undetected, as in *Gladbachus adentatus* [24]. Teeth are inferred to be homologous across gnathostomes [7], demanding numerous independent losses of teeth in stem-chondrichthyans (figures 7 and 8).

*Tooth whorls* (dark green icons, figures 7 and 8) are tooth files with a fused bony base, which grow by the lingual addition of new cusps along a single axis [20]. Tooth whorls are understood to be the evolutionary precursor to modern chondrichthyan tooth families, which have a similar morphology but lack a common bony base [65]. Tooth whorls are present in a range of stem-group chondrichthyans with otherwise dissimilar anatomies, including those with dentigerous jaw bones (e.g. *Ischnacanthus* [22]), densely tesserate head skeletons (e.g. *Climatius* [66]), and more conventionally shark-like taxa (e.g. *Doliodus* [17]). They have also been described in the acanthodiform-like *Latviacanthus* [67], although as this is based on x-ray plates these may be mischaracterized hyoid rakers as in *Homalacanthus* [59]. Tooth whorls are also present in some stem-group holocephalans such as iniopterygians [68], although these are likely to be secondarily derived given their phylogenetic remoteness from the chondrichthyan stem-group. Some osteichthyan taxa also possess tooth whorls, but here the tooth crowns are shed via resorption of the tooth base, a mechanism not present in statodont chondrichthyan tooth whorls [14]. There is variation in their distribution on the jaw: in ischnacanthids and osteichthyans, tooth whorls are few in number and limited to the symphysis [22], whereas in more crownward chondrichthyans they are arrayed along the length of the jaw and comprise the entire dentition (gold and yellow-green icons, figures 7 and 8). The distribution of tooth whorls across osteichthyans and within chondrichthyans is complex, with whorl-bearing taxa often nested within whorl-less radiations. Probabilistic ancestral state reconstruction indicates that tooth whorls evolved independently multiple times both within chondrichthyans and across gnathostomes [7]. Within chondrichthyans, different phylogenetic topologies have quite different implications for the gain and loss of tooth whorls and their distribution across the group (figure 8).

Some chondrichthyans have *teeth that are not organized into files* but which lie directly on the jaw cartilage (coral icons [in part; this icon also captures teeth arranged in files that are borne on the jaw cartilages], figures 7 and 8). This condition is present in *Acanthodopsis* (figure 5), *Pucapampella*, and *Gladbachus* [18,24], although expressed in different ways. In *Acanthodopsis*, teeth are triangular in profile and diminish in size anteriorly and posteriorly (figure 5). In *Pucapampella*, teeth form a single row along the jaw but show a variety of sizes, shapes and spacings [18]. In *Gladbachus*, teeth are much reduced and individually separate, although possibly aligned linguo-labially [24]. However, these taxa are scattered across the tree, and their tooth morphologies can be radically different, suggesting that teeth that lie directly on the jaw cartilage and are not organized into files evolved multiple times independently. Although *Gladbachus* and *Pucapampella* are recovered as a clade in our analyses, this is contrary to most other recent findings (figure 8c,d) and we view this result with extreme caution.

*Dentition cones* are tooth-like cones with smaller denticles attached. They are only known in three partially articulated ischnacanthids (*Zemylacanthus* (*Poracanthodes*), *Acritolepis* and *Serradentus* [54,69,70]), and are absent in the better characterized *Ischnacanthus* and relatives [22,36]. The lack of fully articulated fossils bearing dentition cones leaves open the possibility that they represent a displaced part of the branchial apparatus rather than oral structures [50], and are perhaps comparable to gill and hyoid rakers in cheiracanthids and acanthodids [35,59]. Whether oral or branchial in origin, their presence may unite the subset of ischnacanthids that possess them, although this has not been tested in a phylogenetic context.

*Tooth-like scales* are present along the oral margin of some stem-group chondrichthyans and include part of the cheek squamation, the 'lip' and the ventral rostral area. They are best characterized in *Ischnacanthus*-like 'acanthodians' [36] in which they show a variety of morphologies, and in life may have helped with grasping prey. Some of these scales are strikingly similar in organization to tooth whorls (which are also present within the gape of the same animals), comprising a file of denticles oriented towards the mouth [22,36]. Specialized tooth-like scales have also been identified along the

margin of the mouth in *Obtusacanthus* [62]. More generally, tooth-like denticles are common along the oral margin of the tooth row in early osteichthyans [14,71]. Although potentially interesting from a developmental perspective, they seem unlikely to carry any phylogenetic signal.

*Dentigerous jaw bones* are tooth-bearing dermal jaw bones present in the upper and lower jaws of a number of stem-chondrichthyan taxa [23]. Articulated fossils bearing dentigerous jaw bones include *Zemlyacanthus*, *Nerepisacanthus* and *Serradentus* [32,54,72]. By far the best anatomically characterized taxa with dentigerous jaw bones are *Ischnacanthus* and similar taxa [22,25]. There are few anatomical characters to group taxa possessing dentigerous jaw bones, but all have a complement of oral structures including some combination of symphyseal tooth whorls, dentition cones and tooth-like cheek scales. Dentigerous jaw bones themselves display anatomical diversity, for example relating to the structure of the bone, the number and morphology of tooth rows [23] and variance in dentition shapes likely linked to diet [25]. In our phylogeny, taxa with dentigerous jaw bones (i.e. ischnacanthids) are recovered in a polytomy, in a broader grouping of 'acanthodians' with dermal mouth plates (dark purple icons, figures 7 and 8). Dermal jaw bones, both edentulous and tooth-bearing, are also present in 'placoderms' and osteichthyans, but few phylogenetic results support their homology with those of ischnacanthids (figures 7 and 8).

*Occlusal plates* are a pair of smooth dermal plates in the gapes of some stem-chondrichthyans (light purple icons, figures 7 and 8). Their detailed anatomy is poorly characterized and in the past they have become terminologically and anatomically confused with the mandibular splint [30]. Occlusal plates are present in *Diplacanthus*, *Rhadinacanthus*, *Milesacanthus*, *Uraniacanthus*, *Culmacanthus* and *Tetanopsyrus* [28–30,40,73,74]. At least some of these taxa have other common morphologies (i.e. similar body shapes, scapular processes with posterior lamina, large postorbital scales, deep, striated dorsal fin spine insertions), and on this basis they are grouped into the diplacanthids [28]. There is some variation in the morphology of occlusal plates. In all taxa but *Tetanopsyrus* [29,75], they are only present in the lower jaws. *Tetanopsyrus* may also have tooth-like denticles along the inner surface of the plates, although this is only known from an isolated Meckel's cartilage associated with a complete *Tetanopsyrus* specimen [29] and its attribution is, therefore, uncertain. In *Uraniacanthus* and *Culmacanthus*, a dorsal process is present [40,73]. We recover diplacanthids as monophyletic in our Bayesian analysis (figure 7; electronic supplementary material, figure S3), but paraphyletic with respect to *Tetanopsyrus* in our parsimony analysis, and occlusal plates appear to be a character uniting diplacanthids (figures 7 and 8). *Tetanopsyrus*, with its upper and lower plates, may represent a link between the occlusal plates of diplacanthids and dermal jaw bones of ischnacanthids.

A *mandibular splint* (variously termed dentohyoid, extramandibular spine, splenial or mandibular bone) is a slightly sinusoidal dermal bone that underlies the Meckel's cartilage ventrolaterally (grey icons, figures 7 and 8). Unlike the other structures discussed here, it did not lie within the gape, and likely reinforced the lower jaw. Mandibular splints are present in *Acanthodes*, *Acanthodopsis*, *Halimacanthodes*, *Howittacanthus* and *Protogonacanthus* [56,58,76,77]. They have also been incorrectly identified in a variety of other taxa. Mandibular splints in mesacanthids [34,46] are more similar in size to gular plates and may represent displaced elements of this series. Reports in diplacanthids [73,74] are better interpreted as occlusal plates [30]. Although a mandibular splint has been identified in the putative cheiracanthid *Protogonacanthus* [58], the taxon in question is likely not a cheiracanthid but an acanthodid [59]. Finally, as we show, descriptions of mandibular splints in ischnacanthids [41,50] instead represent a reinforced ventral margin of the endoskeletal mandible. Mandibular splints in acanthodids are very conservative in form, although maybe ornamented as in *Acanthodes sulcatus* [58]. Its similarity to the ventral branchiostegal rays in *Acanthodes* (figure 6a,b), which are also dermal, tubular and slighty sinusoidal, suggests that it may be part of this series that has been co-opted to support the jaw. Our phylogeny suggests that a mandibular splint unites *Acanthodes* and *Acanthodopsis* but either evolved convergently in *Halimacanthodes* (figure 7) or was lost in *Homalacanthus*; we consider it most likely that it unites acanthodids to the exclusion of other stem-chondrichthyans, and that this distribution is a result of undersampling acanthodids and their characters in our phylogeny.

## 4.3. The evolution of chondrichthyan teeth

Although phylogenetic topologies for early chondrichthyans are poorly resolved and often suggest conflicting hypotheses, there are some signals that may provide insight into the evolution of a modern shark-like dentition. The placement of 'climatiid' acanthodians in a relatively crownward position on the chondrichthyan stem [24,32,44,45] (figures 7 and 8a–c) suggests that taxa with a dentition entirely formed from tooth whorls share a last common ancestor to the exclusion of other stem-group

chondrichthyan taxa. Not all topologies support this hypothesis, however: Frey *et al.* [46] recover climatiids as remote from the chondrichthyan crown node, implying that an extensively whorl-based dentition borne on the jaw cartilages either developed independently in the crownward lineage or was lost in the Acanthodii *sensu* Coates *et al.* [24]. This phylogeny is based on a more limited selection of stem-group chondrichthyan taxa, which may have had an influence on reconstructed patterns of character evolution. Either scenario still invokes multiple episodes of secondary tooth loss (e.g. *Lupoposyrus*) and divergences from a whorl-like tooth anatomy (*Gladbachus*, *Pucapampella*). In our phylogeny, tooth shedding is restricted to the crown node. Generative tooth series are present in stem-group elasmobranchs (e.g. *Phoebodus* [38]) and stem-group holocephalans with both shark-like (e.g. *Ferromirum* [46]) and more chimaeroid-like (e.g. *Debeerius* [78]) forms. However, it is unclear how widespread the non-shedding condition described in some sharks with cladodont teeth [79] is. Either way, this suggests a 'two-step' development of the stereotypical chondrichthyan dentition, with an initial shift towards tooth whorls borne exclusively on the jaw cartilages, followed by the eventual loss of fused bases and concomitant development of tooth shedding in crown-group chondrichthyans. However, significant phylogenetic uncertainty persists, and this scenario warrants further testing as hypotheses of relationship stabilize. Despite this, the interposition of multiple lineages of non-shedding stem-chondrichthyan taxa between shedding chondrichthyans and shedding osteichthyans confirms that a shedding dentition evolved twice, in two different ways, in crown-gnathostomes [6,7,11,14].

In stark contrast to the clade comprising chondrichthyans with tooth whorls, the clade or grade including diplacanthids, acanthodids and ischnacanthiforms exhibits a diverse array of oral structures, none of which seem to persist beyond the end of the Palaeozoic [24]. In phylogenetic analyses, this grade is consistently recovered at the base of the chondrichthyan total group, with the exception of Frey *et al.* [46], who recover it in a more crownward position. A number of likely apomorphic oral morphologies are present within this clade, including diplacanthid occlusal plates, ischnacanthid dentigerous jaw bones and toothless acanthodids with mandibular splints. Many of these morphologies are known from the Late Silurian and Devonian, approximately contemporaneously to 'acanthodians' with tooth whorls and the unusual dentitions in more shark-like taxa (e.g. *Pucapampella*, *Gladbachus*). Novel oral morphologies have been linked to a period of inferred rapid gnathostome evolution [7,45]. Furthermore, in the Devonian small-bodied chondrichthyans were significant in freshwater nektonic faunas [24]: diverse oral structures seem likely to have accompanied their radiation into these niches. While the latest surviving lineage of the 'acanthodian' grade was the remarkably morphologically conservative acanthodids [76], *Acanthodopsis* shows that experimentation with novel oral apparatus in stem-group chondrichthyans continued well into the Late Carboniferous.

# 5. Summary

'Acanthodian' stem-group chondrichthyans display a diverse array of oral and dental morphologies, including an apparently independent origin of teeth deep within an edentulous clade. However, interpreting patterns of tooth evolution is complicated by conflicting and unresolved phylogenetic hypotheses, both for the chondrichthyan (figure 7; electronic supplementary material, figures S1 and S2) and gnathostome stem-group (e.g. [6,45]). CT- and synchrotron-based investigations seem likely to provide the anatomical information necessary to resolve these instabilities, and likelihood-based methods provide another potential way of overcoming uncertainties [7]. In the meantime, the proliferation of different tree shapes in conjunction with generally low support values means that morphologies should be considered across multiple potential topologies. This illustrates the challenges of drawing broad-scale conclusions for gnathostome tooth evolution on the basis of unstable relationships or tentatively placed taxa.

Ethics. This research is based exclusively on specimens from natural history collections.
Data accessibility. Raw data (.vol or .tiff stacks), Mimics files and 3D PLY files for each specimen are deposited in Zenodo (doi:10.5281/zenodo.5238205). Data for performing phylogenetic analyses are provided in the electronic supplementary material [80].
Authors' contributions. S.G. conceived the project and selected specimens. S.G. and R.P.D. carried out CT scanning. R.P.D. segmented the specimens, made Blender renders and constructed figures with input from S.G. S.G. and R.P.D. drafted the manuscript. Both authors revised and edited the manuscript, approved the final version and agree to be accountable for all aspects of the work.
Competing interests. We declare we have no competing interests.

Funding. This work was supported by a Junior Research Fellowship, Christ Church, Oxford and a Royal Society Dorothy Hodgkin Research Fellowship, both to S.G. R.P.D. was supported by a Paris Île-de-France Région—DIM 'Matériaux anciens et patrimoniaux' grant (PHARE project).

Acknowledgements. We thank E. Bernard and Z. Johanson (NHMUK) for assistance with specimen access, and V. Fernandez and B. Clark (both NHMUK) and T. Davies (University of Bristol) for assistance with CT scanning. M. Brazeau (Imperial College London), B. Davidson and C. Burrow (Queensland Museum) and P. Ahlberg (Uppsala University) contributed to helpful discussion. M. Colfer (University of Oxford) carried out preliminary segmentation and interpretation of the Mimics files. Four anonymous reviewers provided helpful comments, and we also thank two anonymous reviewers for their comments on an earlier version of this manuscript.

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
