## [Peer Review File · Royal Society Open Science]

Review History

RSOS-210822.R0 (Original submission)

Review form: Reviewer 1

Is the manuscript scientifically sound in its present form?

No

Are the interpretations and conclusions justified by the results?

No

Is the language acceptable?

Yes

Do you have any ethical concerns with this paper?

No

Have you any concerns about statistical analyses in this paper?

No

Recommendation?

Reject

Comments to the Author(s)

Dearden and Giles described the jaw bones and teeth of five acanthodian genera. Then they summarized different oral structures of acanthodians, and mapped these jaw and teeth characters in the recent phylogenetic analyses. The description was based on uCT scans that do not have an adequate resolution for detailed histological studies. The data was presented by surface rendering and virtual thin sections, but no segmentation of internal structures and no 3D insight. Tissues cannot be identified with certainty and the development of the tooth rows cannot be properly comprehended. This makes it especially tricky to interpret the material of *Acanthodopsis*, which is a particularly important taxon in acanthodian dental evolution. This study provides some new anatomical information. However, since articulated specimens of acanthodians are very rare and histology becomes the key to chondrichthyan evolution, the conclusion that can be drawn from this study is limited.

Comments on materials and methods:

The voxel size of the CT scans at 17.3 μm , 19.5 μm , 22.6 μm , 24.6 μm and 44.9 μm is far too large and can provide information no better than the anatomical level. However, to give interpretations at the histological level, which is necessary for a proper understanding and has been tried by the authors, usually requires high-resolution synchrotron scanning with a voxel size lower than 1 μm .

One of the acanthodian genera, *Ischnacanthus*, has been studied by Rücklin et al. 2021, which also using microtomography but with a much higher resolution (synchrotron scan with voxel size 0.74 μm). The authors stated that the teeth lack enameloid, contra the result of Rücklin et al. 2021, which may or may not be true. However, considering the great difference in the resolution of the scans between the two studies, this statement is not convincing: enameloid is unlikely to be visible in the lower-resolution scans of this study. Compared to the classic study of Ørvig 1973, this study is not able to enhance the understanding of acanthodian dentitions to a great degree.

Comments on references and the conclusive figures:

Many contra papers have pointed out that what Rücklin & Donoghue 2015 [ref. 4] described is not a tooth-bearing bone and it is impossible to determine whether it belongs to *Romundina*. Rücklin himself has admitted his mistake in public. However, this manuscript, which does not provide any supporting evidence for it, cited Rücklin & Donoghue 2015 without criticism. This is coupled with the coding of *Romundina* (with teeth) in the matrix, implying the authors still believe Rücklin & Donoghue 2015. Whereas CPW.9, which does certainly have teeth, is definitely NOT *Romundina*. Thus the tree showing the distribution of teeth is misleading.

In the tree of figure 6, stem gnathostomes are only represented by three taxa; actinopterygians and sarcopterygians only have a solo representative, respectively; no stem osteichthyans at all. The “placoderm-osteichthyan transition” is thus not reflected. This has oversimplified the dental evolution of gnathostomes, even though this manuscript focuses on acanthodians. As a consequence, the picture of vertebrate tooth evolution is distorted. Key taxa are not shown, and discussion on the dental diversity of early gnathostomes and their potential evolutionary relationship with the acanthodian dentitions has been dodged. A large number of recent discoveries of stem gnathostomes and stem osteichthyan dentitions are neglected, but only cited as “The advent of micro-computed tomography has led to a renewed interest in tooth evolution and development in Palaeozoic gnathostomes. These have mostly focussed on stem-gnathostome ‘placoderms’ [2,5,6,8] and osteichthyans [9–14].” No comparison of morphological and

histological details, and no discrimination between supported and falsified theories among previous studies, and thus little meaningful contribution to the literature.

A detailed discussion should be made for each icon in figures 6 and 7. Does “x” indicate absent? It should be denoted in the figure legend. What does “teeth ankylosed” mean? The teeth of *Psarolepis* and *Cheirolepis* are definitely ankylosed in the dermal jawbones. Why they are marked as “x”? A tooth whorl is formed by tooth based joined, but each tooth whorl is a separated dermal element, how can the whorl bases joined? What does “whorl bases joined” mean? It does not correspond to any oral structures that are mentioned in Discussion.

The original descriptions that the distribution of the colored icons based on should be cited for each taxon in the figure legend.

Minor comments:

Line 43-44

“lateral to and overlying endoskeletal jaw cartilages” would better correspond to “outer and inner dental arcade”, rather than “inner and outer”.

As in many places throughout the text, “endoskeletal” can be deleted, since there is no exoskeletal cartilages.

Line 46

“labially-directed” should be “labiolingually-directed”.

Line 55-57

Ref. 18 is about the oral scales of jawless fish thelodont, which cannot be assigned to the chondrichthyan total group.

Line 57-60

This long sentence needs to be rephrased. “a staggering array” can be used to described tooth organization, but it seems not the case here and is thus confusing. What is the logic to lump the dermal oral structures together with commas? Teeth on dermal plates (do they mean the dentigerous jaw bones?) or tooth whorls and extramandibular dentitions regard the location and tooth-bearing bones, whereas gracile and molariform regard the tooth shape. References should be cited for the specific dermal oral structure they apply to, for example, ref. 15 should be cited behind “tooth whorls” and “gracile”, ref. 20 should be cited behind “tooth whorls” and “molariform teeth”, etc.

Line 188

Unlike those in the bone, the vascular canals at the tooth bases are not parallel. But to determine whether they are randomly oriented, segmentation and 3D visualization is required. Can they be considered as pulp canals?

Line 189-191

Do the cusps of the medial row also show a size gradient and overlap between cusps? Is the vascularization related to their size?

Line 192

The crown of all teeth in figure 1f looks vascular too. If the crown has fewer vascular canals, is it because they have been infilled?

Line 195-196

In figure 1d, it looks like the tenth cusp is not only smaller than the ninth cusp, but also the seventh cusp.

Line 198

Is the bone around the eighth cusp also damaged?

Line 199

Tomogram sections cutting through the ninth cusp and the neighboring cusps are required.

Line 200

Are the vascular canals of the ninth cusp closely connected to those of the eighth and tenth cusps, and the bone?

Line 201

Does the medial position of the ninth cusp correlate to the most damaged part of the eighth cusp?

Line 206

It seems the ridges are all conspicuously tuberculated in the cusps of the anterior half of the tooth row. Just the number of tubercles depends on the length of the ridges.

Line 207

Does it mean they cannot occlude anteriorly?

Line 244

What is the histological relationship between the two additional cusps and the lingual rows of teeth? Are the two additional cusps similar to those on the medial ridge?

Line 250

Again, the cap looks vascular too. The basal part of the pulp cavity of the younger cusps is more widely open, but vascular canals clearly extend to the crown. The pulp cavities of the older cusps are less widely open, demonstrating infilling is ongoing from the crown to the base as more dentine is deposited centripetally.

Line 305-306

It is very important to show a section through the largest tooth and its anterior and posterior neighbors.

Line 421-423

According to the phylogenetic result, it is impossible to tell whether the taxa with tooth whorls comprising the entire dentition is more crownward than those with tooth whorls limited to the symphysis.

Line 482

How to define dermal mouth plates? If it means large dermal jaw bones, it is, of course, only absent in the chondrichthyans without dentigerous jaw bones.

Line 486-487

Does "occlusal plates" correspond to the "dermal plate smooth" in the figures? If so, they should be referred to in the same way.

Line 489-491

Not all the references cited here have ever mentioned occlusal plates or mandibular splint. It is necessary to denote what the occlusal plates were called in the original description of these taxa. It will be better to make a drawing of a generalized acanthodian head above figure 6 to explain the difference between occlusal plates and mandibular splint by showing their location in the

mouth. Otherwise, it is confusing, because none of the taxa described in this manuscript has occlusal plates.

Line 499

Should “already” be “almost”?

Line 555-557

“a single direction in isolated files on an underlying dermal plate” is a common feature of vertebrate dentitions and one of the basic criteria to recognize true teeth. It by no means supports the comparability between dentigerous jaw bones and tooth whorls.

Line 561-565

What does “isolated files” mean? If “isolated” means each file grows on a separated bone, or if “files” indicates a labiolingual organization of teeth, then the dentigerous jaw bones, which bear multiple anteroposterior tooth rows, do not have isolated files. It is impossible to unite the dental development of dentigerous jaw bones and tooth whorls in this way. It is not uncommon for a tooth whorl to bear three files. Multiple non-shedding tooth files have been seen in acanthothoracid marginal jawbones, and the inter-file spacing is very clear too if that is what “isolated” means. Multiple shedding tooth files are also found in stem osteichthyan marginal jaw bones and tooth cushions, as well as sarcopterygian parasymphysial tooth whorls. Sauropods could generate teeth so quickly that their tooth families appear as the tooth files of modern sharks, while the tooth families of hadrosaurid dental battery are comparable with the chondrichthyan tooth whorls or files of tooth retention. Therefore the lingual tooth addition represented by tooth whorls is likely a primitive setting that is shared by the basal members of different groups of gnathostomes, and with modification, it has been carried on by osteichthyans. The anterior tooth addition is unique to the acanthodian dentigerous jaw bones. Nevertheless, the tooth rows on the dentigerous jaw bones are more comparable to other dentitions with a radial organization, such as in arthrodires and lungfish. All these growth patterns are related to the developmental relationship between odontodes and the underlying bones, and thus similar patterns can appear again and again. Hence, it is more important to explore the developmental mechanism than only extract phylogenetic signals. All in all, isolated files can never be a character uniting chondrichthyans.

Line 592-593

Dermal jaw bones are not likely to present in taxa least proximate to the chondrichthyan crown either.

Line 595-597

There is not enough evidence of the first step that a dentition with both dentigerous jaw bones and tooth whorls shifts to that with tooth whorls only. In the phylogenetic analysis, dentigerous jaw bones only occur in a clade of acanthodians, which may indicate specialization. For the second step, please read Johanson et al. 2020 Integr. Comp.

Line 599

Frey et al. 2020, as the tree in figure 7d, does support a reverse shift that tooth whorls occupy the full length of the jaw in basal acanthodians, and are then partially replaced by dentigerous jaw bones in the clade of ischnacanthids.

Line 603-609

Here the discussion suddenly shifts to the evolutionary relationship with osteichthyan jaw bones and teeth, and the conclusion seems to be that the dental system of acanthodians and osteichthyans evolve completely independently. However, the authors ignored the fact that the teeth of stem osteichthyans are commonly organized in the same way as the chondrichthyan

dentitions entirely made up of tooth whorls, just the whorl bases joined into a large dermal bone. "Whorl bases joined" is exactly what the dark blue irons in figures 6 and 7 indicate. This may be convergent but is well worth mentioning.

Line 613-623

That is right. Therefore, the authors should focus on the actual comparative morphologies, respect any conflicting morphological data, rather than be eager to make up hypotheses by playing with the trees.

Line 617

The dentitions described in ref. 6 are not at all difficult to interpret. The histology is perfectly preserved in 3D and more fully documented than the acanthodian material in this manuscript. They are just difficult to fit in the conventional/the authors' own hypothesis. But just because of that, these dentitions should not be dismissed to avoid addressing the challenges posed by the authors. A detailed comparison with all these dentitions is supposed to be made in the Discussion.

Review form: Reviewer 2

Is the manuscript scientifically sound in its present form?

Yes

Are the interpretations and conclusions justified by the results?

Yes

Is the language acceptable?

Yes

Do you have any ethical concerns with this paper?

No

Have you any concerns about statistical analyses in this paper?

No

Recommendation?

Accept with minor revision (please list in comments)

Comments to the Author(s)

The authors provided a great study and review of oral structures, especially teeth in early chondrichthyans. Great data set, neat method, nice figures, and appropriate analyses. I feel that the paper would benefit from either more taxa labels in figure 6 or clear definitions of group memberships. Given the instability of some taxa and groups it might be easier for the reader if groups are made more clear. I enjoyed the discussion

Other suggestions:

Add posterior probabilities to figure 6

Line 534-537 -> but with low support. Which seems common for the data set... is this ok?

Review form: Reviewer 3

Is the manuscript scientifically sound in its present form?

Yes

Are the interpretations and conclusions justified by the results?

Yes

Is the language acceptable?

Yes

Do you have any ethical concerns with this paper?

No

Have you any concerns about statistical analyses in this paper?

No

Recommendation?

Accept with minor revision (please list in comments)

Comments to the Author(s)

I really enjoyed the CT scans and anatomical discussion of teeth and teeth-like structures presented in your manuscript. The discussion is well-written and explains the different types of teeth structures very well and for the first time clearly illustrates the evolution of teeth, teeth-like structures, and tooth-bearing bones in these groups. My main issues lie in the introduction which is incredibly short. Please find detailed comments in the attached document.

Review form: Reviewer 4

Is the manuscript scientifically sound in its present form?

Yes

Are the interpretations and conclusions justified by the results?

Yes

Is the language acceptable?

Yes

Do you have any ethical concerns with this paper?

No

Have you any concerns about statistical analyses in this paper?

No

Recommendation?

Accept with minor revision (please list in comments)

Comments to the Author(s)

General comments

This is a valuable review and analysis of early chondrichthyan ‘acanthodian’ dentitions, with emphasis placed squarely on *Acanthodopsis* and ischnacanthids.

As a result, the present article needs to distinguish itself clearly from Rucklin et al. 2021 “Acanthodian dental development and the origin of gnathostome dentitions” (nat. ecol. & evol. Reference #7).

Both works focus on ischnacanthid jaw bones and dentitions; both use computed tomography to investigate hard tissue histology and infer patterns of growth; both present hypotheses concerning the evolution of dentitions in early jawed vertebrates from a stem-chondrichthyan perspective. However, Rucklin and colleagues are concerned with teeth in the last common ancestor of crown gnathostomes whereas the present work concentrates on dental diversity in stem-chondrichthyans. The takeaway message from the present work is that despite noise in the data set, *Acanthodopsis* is confirmed as an acanthodid (not an ischnacanthid), and, as such, evolved its teeth independently within an otherwise toothless clade. Interesting!

For these reasons, I recommend retaining the guts of the paper - data, descriptions, analyses and results, but shift the emphasis in the discussion and intro.; changing the title and abstract accordingly.

Clearer images of the CT histology would help – perhaps line-drawn diagrams depicting where the authors identify boundaries between tissues?

Might the addition of a comparative figure aid the descriptions and discussion?

Detailed comments

Abstract

Re-draft with greater emphasis on new findings and how these change or challenge current views.

L. 56-57 – list the three (?) stem-chondrichthyans described using CT: *Doliodus*, *Acanthodes*, *Pucapampella*, ischnacanthids (various), *Gladbachus*...

Stem chondrichthyan dentitions are varied, but ‘staggering’ (as a superlative) is a bit strong compared with, for example, mammal teeth running the gamut from narwhales to elephants, rodents, aardvarks, crab eating seals, and the products of US dentistry.

L. 81 – every hypothesis deserves a reference: Long ‘86?

L. 88-92 – to be clear, *Atopacanthus* is known only from disarticulated remains?

L.100 & elsewhere... lots of elsewhere... try not to overuse ‘element’. If a more informative word such as ‘bone’ or ‘cartilage’ could be substituted, then do so.

L.167 – suggestion – the circular ridge (*Taemasacanthus*) seems far more likely to mark an attachment for a labial ligament.

L. 186 – Fig. 1 part ‘d’ – colour code the teeth consistently with parts ‘a’ and ‘b’. In the present version there appear to be only two rows of teeth: lateral and lingual. Teeth on the mesial ridge are barely visible. Italicize taxon names in caption – here & elsewhere.

L. 209-210. Perichondral tissue – is this bone?

L. 217 – what is a typical (although tipler) ischnacanthid process? And, according to whom is this process characteristic (uniquely?) of ischnacanthids? Burrow? Long?

L. 235 – Fig.2 part ‘d’ colour code the teeth consistently with parts ‘a’ and ‘b’ (again).

Add caption detail to identify anterior (distal) and posterior (proximal) ends of bone.
Add reference to the insert diagram of possible bone locations in jaws.

L. 259. No medial ridge identified in the figure

L. 260. Only three cusps visible – and a large crack where a fourth might be?

L. 301. In the figure, the teeth are not histologically distinct from the bone forming the perichondral sheath of Meckel's cartilage. This might be an artefact of pdf figure quality (enlarging the image doesn't improve resolution).

Perhaps a line diagram showing the authors' interpretation of the CT histology is needed?

L. 306. I don't see the overlap between teeth.

L. 391. Erratic use of bold text.

L. 407, 433, 447, 456, 468, 486, 507 – bold text – are these subheadings? Some at start of lines; some embedded within sentences?

L. 398. Possible exception? Are there reasons to challenge the teeth described in *Tetanopsyrus* (ref.24)?

L. 433. Paragraph starting 'Some chondrichthyans have teeth that are not organized into files but...' – muddled structure. This section seems to be about teeth that lie directly on the jaw cartilage – file organization seems secondary. Might be worth rewriting.

L. 566-568. At last! Here's the pay-off, but it's buried deep in the discussion.

L. 575. Not quite sure about the meaning of 'likely dermal in origin': explain.

L. 596. 'borne' rather than 'bone'.

L.707. Ref. 7 - add doi: [10.1038/s41559-021-01458-4](https://doi.org/10.1038/s41559-021-01458-4)

L. 760 Ref. 23 - add the correct doi -this one links to a different ref. (*Euthacanthus*)

Decision letter (RSOS-210822.R0)

Dear Dr Giles

The Editors assigned to your paper RSOS-210822 "Tomographic data of 'acanthodian' oral structures and a review of dental diversity in early chondrichthyans" have now received comments from reviewers and would like you to revise the paper in accordance with the reviewer comments and any comments from the Editors. Please note this decision does not guarantee eventual acceptance.

We invite you to respond to the comments supplied below and revise your manuscript. Below the referees' and Editors' comments (where applicable) we provide additional requirements.

Final acceptance of your manuscript is dependent on these requirements being met. We provide guidance below to help you prepare your revision.

You will see that whilst reviewers 2, 3 and 4 make positive comment about the paper (though they all recommend changes prior to publication), reviewer 1 is much more negative. Please pay particular attention to the comments of reviewer 1 in your responses and revisions.

Please submit your revised manuscript and required files (see below) no later than 21 days from today's (ie 02-Aug-2021) date. Note: the ScholarOne system will 'lock' if submission of the revision is attempted 21 or more days after the deadline. If you do not think you will be able to meet this deadline please contact the editorial office immediately.

on behalf of Professor Peter Haynes (Subject Editor)
openscience@royalsociety.org

Associate Editor Comments to Author:

An unusually large number of reviewers have been kind enough to return reports on your paper. While the overall picture appears to be positive, there are enough comments, queries and suggestions that we would like you to revise the manuscript as far as possible to take these concerns into consideration. Good luck!

Reviewer comments to Author:

Reviewer: 1

Comments to the Author(s)

Dearden and Giles described the jaw bones and teeth of five acanthodian genera. Then they summarized different oral structures of acanthodians, and mapped these jaw and teeth characters in the recent phylogenetic analyses. The description was based on uCT scans that do not have an adequate resolution for detailed histological studies. The data was presented by surface rendering and virtual thin sections, but no segmentation of internal structures and no 3D insight. Tissues cannot be identified with certainty and the development of the tooth rows cannot be properly comprehended. This makes it especially tricky to interpret the material of *Acanthodopsis*, which is a particularly important taxon in acanthodian dental evolution. This study provides

some new anatomical information. However, since articulated specimens of acanthodians are very rare and histology becomes the key to chondrichthyan evolution, the conclusion that can be drawn from this study is limited.

Comments on materials and methods:

The voxel size of the CT scans at 17.3 μm , 19.5 μm , 22.6 μm , 24.6 μm and 44.9 μm is far too large and can provide information no better than the anatomical level. However, to give interpretations at the histological level, which is necessary for a proper understanding and has been tried by the authors, usually requires high-resolution synchrotron scanning with a voxel size lower than 1 μm .

One of the acanthodian genera, *Ischnacanthus*, has been studied by Rücklin et al. 2021, which also using microtomography but with a much higher resolution (synchrotron scan with voxel size 0.74 μm). The authors stated that the teeth lack enameloid, contra the result of Rücklin et al. 2021, which may or may not be true. However, considering the great difference in the resolution of the scans between the two studies, this statement is not convincing: enameloid is unlikely to be visible in the lower-resolution scans of this study. Compared to the classic study of Ørvig 1973, this study is not able to enhance the understanding of acanthodian dentitions to a great degree.

Comments on references and the conclusive figures:

Many contra papers have pointed out that what Rücklin & Donoghue 2015 [ref. 4] described is not a tooth-bearing bone and it is impossible to determine whether it belongs to *Romundina*. Rücklin himself has admitted his mistake in public. However, this manuscript, which does not provide any supporting evidence for it, cited Rücklin & Donoghue 2015 without criticism. This is coupled with the coding of *Romundina* (with teeth) in the matrix, implying the authors still believe Rücklin & Donoghue 2015. Whereas CPW.9, which does certainly have teeth, is definitely NOT *Romundina*. Thus the tree showing the distribution of teeth is misleading.

In the tree of figure 6, stem gnathostomes are only represented by three taxa; actinopterygians and sarcopterygians only have a solo representative, respectively; no stem osteichthyans at all. The “placoderm-osteichthyan transition” is thus not reflected. This has oversimplified the dental evolution of gnathostomes, even though this manuscript focuses on acanthodians. As a consequence, the picture of vertebrate tooth evolution is distorted. Key taxa are not shown, and discussion on the dental diversity of early gnathostomes and their potential evolutionary relationship with the acanthodian dentitions has been dodged. A large number of recent discoveries of stem gnathostomes and stem osteichthyan dentitions are neglected, but only cited as “The advent of micro-computed tomography has led to a renewed interest in tooth evolution and development in Palaeozoic gnathostomes. These have mostly focussed on stem-gnathostome ‘placoderms’ [2,5,6,8] and osteichthyans [9-14].” No comparison of morphological and histological details, and no discrimination between supported and falsified theories among previous studies, and thus little meaningful contribution to the literature.

A detailed discussion should be made for each icon in figures 6 and 7. Does “x” indicate absent? It should be denoted in the figure legend. What does “teeth ankylosed” mean? The teeth of *Psarolepis* and *Cheirolepis* are definitely ankylosed in the dermal jawbones. Why they are marked as “x”? A tooth whorl is formed by tooth based joined, but each tooth whorl is a separated dermal element, how can the whorl bases joined? What does “whorl bases joined” mean? It does not correspond to any oral structures that are mentioned in Discussion.

The original descriptions that the distribution of the colored icons based on should be cited for each taxon in the figure legend.

Minor comments:

Line 43-44

“lateral to and overlying endoskeletal jaw cartilages” would better correspond to “outer and inner dental arcade”, rather than “inner and outer”.

As in many places throughout the text, “endoskeletal” can be deleted, since there is no exoskeletal cartilages.

Line 46

“labially-directed” should be “labiolingually-directed”.

Line 55-57

Ref. 18 is about the oral scales of jawless fish thelodont, which cannot be assigned to the chondrichthyan total group.

Line 57-60

This long sentence needs to be rephrased. “a staggering array” can be used to describe tooth organization, but it seems not the case here and is thus confusing. What is the logic to lump the dermal oral structures together with commas? Teeth on dermal plates (do they mean the dentigerous jaw bones?) or tooth whorls and extramandibular dentitions regard the location and tooth-bearing bones, whereas gracile and molariform regard the tooth shape. References should be cited for the specific dermal oral structure they apply to, for example, ref. 15 should be cited behind “tooth whorls” and “gracile”, ref. 20 should be cited behind “tooth whorls” and “molariform teeth”, etc.

Line 188

Unlike those in the bone, the vascular canals at the tooth bases are not parallel. But to determine whether they are randomly oriented, segmentation and 3D visualization is required. Can they be considered as pulp canals?

Line 189-191

Do the cusps of the medial row also show a size gradient and overlap between cusps? Is the vascularization related to their size?

Line 192

The crown of all teeth in figure 1f looks vascular too. If the crown has fewer vascular canals, is it because they have been infilled?

Line 195-196

In figure 1d, it looks like the tenth cusp is not only smaller than the ninth cusp, but also the seventh cusp.

Line 198

Is the bone around the eighth cusp also damaged?

Line 199

Tomogram sections cutting through the ninth cusp and the neighboring cusps are required.

Line 200

Are the vascular canals of the ninth cusp closely connected to those of the eighth and tenth cusps, and the bone?

Line 201

Does the medial position of the ninth cusp correlate to the most damaged part of the eighth cusp?

Line 206

It seems the ridges are all conspicuously tuberculated in the cusps of the anterior half of the tooth row. Just the number of tubercles depends on the length of the ridges.

Line 207

Does it mean they cannot occlude anteriorly?

Line 244

What is the histological relationship between the two additional cusps and the lingual rows of teeth? Are the two additional cusps similar to those on the medial ridge?

Line 250

Again, the cap looks vascular too. The basal part of the pulp cavity of the younger cusps is more widely open, but vascular canals clearly extend to the crown. The pulp cavities of the older cusps are less widely open, demonstrating infilling is ongoing from the crown to the base as more dentine is deposited centripetally.

Line 305-306

It is very important to show a section through the largest tooth and its anterior and posterior neighbors.

Line 421-423

According to the phylogenetic result, it is impossible to tell whether the taxa with tooth whorls comprising the entire dentition is more crownward than those with tooth whorls limited to the symphysis.

Line 482

How to define dermal mouth plates? If it means large dermal jaw bones, it is, of course, only absent in the chondrichthyans without dentigerous jaw bones.

Line 486-487

Does "occlusal plates" correspond to the "dermal plate smooth" in the figures? If so, they should be referred to in the same way.

Line 489-491

Not all the references cited here have ever mentioned occlusal plates or mandibular splint. It is necessary to denote what the occlusal plates were called in the original description of these taxa. It will be better to make a drawing of a generalized acanthodian head above figure 6 to explain the difference between occlusal plates and mandibular splint by showing their location in the mouth. Otherwise, it is confusing, because none of the taxa described in this manuscript has occlusal plates.

Line 499

Should "already" be "almost"?

Line 555-557

"a single direction in isolated files on an underlying dermal plate" is a common feature of vertebrate dentitions and one of the basic criteria to recognize true teeth. It by no means supports the comparability between dentigerous jaw bones and tooth whorls.

Line 561-565

What does “isolated files” mean? If “isolated” means each file grows on a separated bone, or if “files” indicates a labiolingual organization of teeth, then the dentigerous jaw bones, which bear multiple anteroposterior tooth rows, do not have isolated files. It is impossible to unite the dental development of dentigerous jaw bones and tooth whorls in this way. It is not uncommon for a tooth whorl to bear three files. Multiple non-shedding tooth files have been seen in acanthothoracid marginal jawbones, and the inter-file spacing is very clear too if that is what “isolated” means. Multiple shedding tooth files are also found in stem osteichthyan marginal jaw bones and tooth cushions, as well as sarcopterygian parasymphysial tooth whorls. Sauropods could generate teeth so quickly that their tooth families appear as the tooth files of modern sharks, while the tooth families of hadrosaurid dental battery are comparable with the chondrichthyan tooth whorls or files of tooth retention. Therefore the lingual tooth addition represented by tooth whorls is likely a primitive setting that is shared by the basal members of different groups of gnathostomes, and with modification, it has been carried on by osteichthyans. The anterior tooth addition is unique to the acanthodian dentigerous jaw bones. Nevertheless, the tooth rows on the dentigerous jaw bones are more comparable to other dentitions with a radial organization, such as in arthrodires and lungfish. All these growth patterns are related to the developmental relationship between odontodes and the underlying bones, and thus similar patterns can appear again and again. Hence, it is more important to explore the developmental mechanism than only extract phylogenetic signals. All in all, isolated files can never be a character uniting chondrichthyans.

Line 592-593

Dermal jaw bones are not likely to present in taxa least proximate to the chondrichthyan crown either.

Line 595-597

There is not enough evidence of the first step that a dentition with both dentigerous jaw bones and tooth whorls shifts to that with tooth whorls only. In the phylogenetic analysis, dentigerous jaw bones only occur in a clade of acanthodians, which may indicate specialization. For the second step, please read Johanson et al. 2020 Integr. Comp.

Line 599

Frey et al. 2020, as the tree in figure 7d, does support a reverse shift that tooth whorls occupy the full length of the jaw in basal acanthodians, and are then partially replaced by dentigerous jaw bones in the clade of ischnacanthids.

Line 603-609

Here the discussion suddenly shifts to the evolutionary relationship with osteichthyan jaw bones and teeth, and the conclusion seems to be that the dental system of acanthodians and osteichthyans evolve completely independently. However, the authors ignored the fact that the teeth of stem osteichthyans are commonly organized in the same way as the chondrichthyan dentitions entirely made up of tooth whorls, just the whorl bases joined into a large dermal bone. “Whorl bases joined” is exactly what the dark blue irons in figures 6 and 7 indicate. This may be convergent but is well worth mentioning.

Line 613-623

That is right. Therefore, the authors should focus on the actual comparative morphologies, respect any conflicting morphological data, rather than be eager to make up hypotheses by playing with the trees.

Line 617

The dentitions described in ref. 6 are not at all difficult to interpret. The histology is perfectly preserved in 3D and more fully documented than the acanthodian material in this manuscript.

They are just difficult to fit in the conventional/the authors' own hypothesis. But just because of that, these dentitions should not be dismissed to avoid addressing the challenges posed by the authors. A detailed comparison with all these dentitions is supposed to be made in the Discussion.

Reviewer: 2

Comments to the Author(s)

The authors provided a great study and review of oral structures, especially teeth in early chondrichthyans. Great data set, neat method, nice figures, and appropriate analyses. I feel that the paper would benefit from either more taxa labels in figure 6 or clear definitions of group memberships. Given the instability of some taxa and groups it might be easier for the reader if groups are made more clear. I enjoyed the discussion

Other suggestions:

Add posterior probabilities to figure 6

Line 534-537 -> but with low support. Which seems common for the data set... is this ok?

Reviewer: 3

Comments to the Author(s)

I really enjoyed the CT scans and anatomical discussion of teeth and teeth-like structures presented in your manuscript. The discussion is well-written and explains the different types of teeth structures very well and for the first time clearly illustrates the evolution of teeth, teeth-like structures, and tooth-bearing bones in these groups. My main issues lie in the introduction which is incredibly short. Please find detailed comments in the attached document ("RSOS-210822 review-submit.pdf").

Reviewer: 4

Comments to the Author(s)

General comments

This is a valuable review and analysis of early chondrichthyan 'acanthodian' dentitions, with emphasis placed squarely on Acanthodopsis and ischnacanthids.

As a result, the present article needs to distinguish itself clearly from Rucklin et al. 2021

"Acanthodian dental development and the origin of gnathostome dentitions" (nat. ecol. & evol. Reference #7).

Both works focus on ischnacanthid jaw bones and dentitions; both use computed tomography to investigate hard tissue histology and infer patterns of growth; both present hypotheses concerning the evolution of dentitions in early jawed vertebrates from a stem-chondrichthyan perspective. However, Rucklin and colleagues are concerned with teeth in the last common ancestor of crown gnathostomes whereas the present work concentrates on dental diversity in stem-chondrichthyans. The takeaway message from the present work is that despite noise in the data set, Acanthodopsis is confirmed as an acanthodid (not an ischnacanthid), and, as such, evolved its teeth independently within an otherwise toothless clade. Interesting!

For these reasons, I recommend retaining the guts of the paper - data, descriptions, analyses and results, but shift the emphasis in the discussion and intro.; changing the title and abstract accordingly.

Clearer images of the CT histology would help - perhaps line-drawn diagrams depicting where the authors identify boundaries between tissues?

Might the addition of a comparative figure aid the descriptions and discussion?

Detailed comments

Abstract

Re-draft with greater emphasis on new findings and how these change or challenge current views.

L. 56-57 - list the three (?) stem-chondrichthyans described using CT: *Doliodus*, *Acanthodes*, *Pucapampella*, ischnacanthids (various), *Gladbachus*...

Stem chondrichthyan dentitions are varied, but 'staggering' (as a superlative) is a bit strong compared with, for example, mammal teeth running the gamut from narwhales to elephants, rodents, aardvarks, crab eating seals, and the products of US dentistry.

L. 81 - every hypothesis deserves a reference: Long '86?

L. 88-92 - to be clear, *Atopacanthus* is known only from disarticulated remains?

L.100 & elsewhere... lots of elsewhere... try not to overuse 'element'. If a more informative word such as 'bone' or 'cartilage' could be substituted, then do so.

L.167 - suggestion - the circular ridge (*Taemasacanthus*) seems far more likely to mark an attachment for a labial ligament.

L. 186 - Fig. 1 part 'd' - colour code the teeth consistently with parts 'a' and 'b'. In the present version there appear to be only two rows of teeth: lateral and lingual. Teeth on the mesial ridge are barely visible. Italicize taxon names in caption - here & elsewhere.

L. 209-210. Perichondral tissue - is this bone?

L. 217 - what is a typical (although tipless) ischnacanthid process? And, according to whom is this process characteristic (uniquely?) of ischnacanthids? Burrow? Long?

L. 235 - Fig.2 part 'd' colour code the teeth consistently with parts 'a' and 'b' (again). Add caption detail to identify anterior (distal) and posterior (proximal) ends of bone. Add reference to the insert diagram of possible bone locations in jaws.

L. 259. No medial ridge identified in the figure

L. 260. Only three cusps visible - and a large crack where a fourth might be?

L. 301. In the figure, the teeth are not histologically distinct from the bone forming the perichondral sheath of Meckel's cartilage. This might be an artefact of pdf figure quality (enlarging the image doesn't improve resolution). Perhaps a line diagram showing the authors' interpretation of the CT histology is needed?

L. 306. I don't see the overlap between teeth.

L. 391. Erratic use of bold text.

L. 407, 433, 447, 456, 468, 486, 507 - bold text - are these subheadings? Some at start of lines; some embedded within sentences?

L. 398. Possible exception? Are there reasons to challenge the teeth described in *Tetanopsyrus* (ref.24)?

L. 433. Paragraph starting 'Some chondrichthyans have teeth that are not organized into files but...' - muddled structure. This section seems to be about teeth that lie directly on the jaw cartilage - file organization seems secondary. Might be worth rewriting.

L. 566-568. At last! Here's the pay-off, but it's buried deep in the discussion.

L. 575. Not quite sure about the meaning of 'likely dermal in origin': explain.

L. 596. 'borne' rather than 'bone'.

L.707. Ref. 7 - add doi: 10.1038/s41559-021-01458-4

L. 760 Ref. 23 - add the correct doi -this one links to a different ref. (Euthacanthus)

===PREPARING YOUR MANUSCRIPT===

===PREPARING YOUR REVISION IN SCHOLARONE===

Author's Response to Decision Letter for (RSOS-210822.R0)

See Appendix A.

RSOS-210822.R1 (Revision)

Review form: Reviewer 1

Is the manuscript scientifically sound in its present form?

No

Are the interpretations and conclusions justified by the results?

No

Is the language acceptable?

No

Do you have any ethical concerns with this paper?

No

Have you any concerns about statistical analyses in this paper?

No

Recommendation?

Major revision is needed (please make suggestions in comments)

Comments to the Author(s)

The most harmful problem of this manuscript is that the authors refuse to present a complete picture of what has been known about early gnathostome dentitions, but only selected information has been shown. For instance, they clearly don't want to take on board the significance of the transverse organisation (file-like arrangement) of the dentitions in stem osteichthyans and acanthothoracid stem gnathostomes, which is shared with chondrichthyan tooth whorls and tooth files, but distinct from the acanthodian dentigerous jaw bones. Consequently, the authors produce a false image that tooth whorls and dentigerous jaw bones can be united as a synapomorphy of chondrichthyans, based on the characters that are in fact general to the gnathostome total group. This will distort the understanding of dental evolution. Although this manuscript focuses on acanthodians, a much more serious (though can be brief) review of various early gnathostome dentitions is expected in the Introduction/Discussion. The authors should not just cite a bunch of papers and then run away, saying that other taxa or oral structures are poorly understood, poorly characterized, etc.

Line 21

"parallel, continuously replacing" is not very informative. It means nothing for people that are not familiar with shark dentitions. I suggest you clarify the transverse organisation of the files and the conveyer-belt replacement.

Line 22

the site-specific replacement?

Line 54

What do you mean by "partially" here?

Line 62-64

The dentitions of stem gnathostomes are definitely relevant to the evolution of teeth, no matter how “strange” they are. They are not poorly characterized, and we do have understood quite a bit.

Line 77

This is not an accurate description of the early osteichthyan dentitions.

Line 90-91

The long history of file-like arrangement can be traced back to the basal jawed stem gnathostomes. The file-like arrangement also occurs in stem osteichthyans, rather than limited to chondrichthyans.

Line 94-95

at the labial/outer jaw margin

Line 98-101

“file-like teeth” and “file-like whorls” sound strange. Does the former mean files of teeth that are not fused into a whorl (non-joined teeth), or does it includes the whorl condition? “File-like teeth” sounds like a description of tooth shape. Whereas all tooth whorls bear tooth files, in other words, there are no “non-file-like whorls”. Also, there is an intermediate condition that the teeth are not fused by a common base, but post-functional teeth are retained and packed at the outer margin of the mouth, which looks like a tooth whorl. Shedding or non-shedding is a key difference of the different conditions of file-like organization, and thus should be mentioned in the description of the different conditions. Yes, it has been mentioned in the following text, but it just makes it more confusing for readers who are not familiar with early chondrichthyan dentitions. So far, not a simple definition of tooth whorl has been given, but readers should not be supposed to know what is a tooth whorl.

Line 106

Not sure which mode of tooth replacement you are talking about. Not understand why the topic of file-like organization suddenly becomes the mode of tooth replacement, and then jutmp to tooth whorls in the next sentence. As no osteichthyan is mentioned in this paragraph, there should not be a second mode of tooth replacement, but conveyer-belt style tooth replacement.

Line 120-121

It is also unclear whether tooth whorls are shed from time to time.

Line 132-135

This mode of growth has long been well understood since Ørvig 1973, which is based on several identified taxa. Then Smith 2003. This statement is acceptable for Rücklin et al. 2021 only if say “have been shown by microtomography”. Since this is a common mode of tooth addition in acanthodian dentigerous jaw bones, it is not that important whether the specimens are identified or not.

Rephrase “wider construction”.

Line 137-138

Strange sentence. Are you trying to say “teeth that are not borne by dentigerous jaw bones”? “not part of ... tooth files” is redundant, when saying “not arranged into files”.

Line 228

“gnathal plates” is usually used in placoderms. It is better to stick to “dentigerous jaw bones” for ischnacanthids, or just say “jaw bones” for acanthodians regardless it is dentigerous or not.

Line 264-265

"Elgin" is not a known Middle Devonian locality. In fact, there are no Middle Devonian localities at all around Elgin, only Late Devonian ones such as Scaat Craig and Rosebrae. I wonder where this specimen is from.

Line 368-371

Citation of figures is needed. Why does its position suggest an artefact of growth?

Line 381-383

Shown in any figures?

Line 436-437

Why is it flatter but taller?

Line 463

Is the inner tooth row the same as the lingual tooth row? If so, keep the consistency of the terms. Otherwise, do you mean the inner one of the two rows borne by the medial ridge?

Line 465-468

Label the two additional small cusps in the figures. Do you mean they form an additional tooth row?

Line 550-552

Bad sentence construction.

Line 641-642

Do you mean the radial vasculature? Is the point, which the vasculature outwards from, the ossification centre of both the dermal and endoskeletal bones? Make it clearer with shorter sentences.

Line 662-675

The chondrichthyan dentitions with tooth whorls lining along the length of jaw and whorl-like cheek scales are not site-specific either, as in acanthothoracid stem-gnathostomes and early osteichthyans. All those suggest common patterning mechanisms across early gnathostomes, rather than an apomorphic for chondrichthyans. In other words, many other dental elements across gnathostomes can be regarded as "stretched out" tooth whorls, with new teeth added in various orientations, such as lungfish tooth plates.

It is necessary to define "site-specific" here. Is the site relative to the whole jaw or mouth, or to a dermal plate or an element that unite the teeth? It is especially confusing because you also use "site-specific" to describe the dentitions of bony fishes in the abstract.

You are trying to find a dental character to unite the chondrichthyans. It is nice if there are any.

But if not, don't make it up. It is not helpful to understand early gnathostome evolution. As you said "Support values aside from the chondrichthyan total group and crown-group nodes are typically low", in addition to the situation that the osteichthyan-chondrichthyan node is difficult to place among stem-gnathostomes, it is unwise to put things into boxes and seal the boxes. Early vertebrate phylogeny should be based on morphological observations, instead, your interpretation of the morphology is determined by the phylogeny, which is very unstable.

Although all acanthodians are considered stem chondrichthyans at the moment, the diversity of early chondrichthyan dentitions is supposed to cast light on the potential evolutionary relationship between acanthodians and other early gnathostome groups. If acanthodians, at least some of them, were placed in the osteichthyan stem, as in the old times, do you still think that the dentigerous jaw bones are just a modified form of tooth whorls, or more similar to tooth whorls than any other kinds of gnathostome dentitions? In terms of "site-specific", the linear tooth rows

of osteichthyans are more comparable, such as the row of fang pairs, the row of marginal large teeth and the row of accessory small teeth are all site-specifically arranged. Developmentally, it does not make sense for the dentigerous jaw bones, a kind of “antero-posteriorly ‘stretched out’ tooth whorls” in your words, to evolve from the labio-lingual tooth whorls.

Line 735-736

The tooth whorls of iniopterygians is more likely a retained primitive character, because many other stem holocephalans also have tooth whorls along the jaw.

Line 872-879

Do you mean the dentition entirely formed from tooth whorls share a last common ancestor and develop independently in the crownward lineage is one of the scenarios? It is confusing what scenario Frey et al. suggested, according to your statement. As one of the major types of dentitions in Acanthodii, why do you say it “being lost in Acanthodii”?

Line 896-899

It will be nice to indicate the clade only with tooth whorls and the clade with a diverse array of oral structures at the nodes of the trees if they do form different clades.

Figure 1

In (a), it looks like each tooth is a “tooth file”. It is better to box the file of teeth and label the box with a single line. It will be nice to have an inset of the lateral view just like in (b) for comparison. Abbreviations should be explained in the caption. But it is better not to use abbreviations at all since there is enough space, and the consistency should be kept (“tooth file”, “tooth whorl” etc. are not abbreviated).

Supplementary figure 1

Which taxon does (d) belong to, Taemasacanthus or Atopacanthus? No need to correspond the taxa with the figures twice in the caption, which actually brings up a conflict. (a) and (d) can be both medial views if they belong to the same taxon. (b) is from the box of (a), rather than the lateral view, so all the following figures are wrong. Same for (f). Check the figure citations in the main text. It is also unclear which figures share the same scale bar. Since all taxa are given the same colour, it is better to divide the figures into sections for each taxon by adding some separating lines and labelling each division with the taxon names.

Can the circular ridge be seen in any of the figures? If so, please label it.

Review form: Reviewer 4

Is the manuscript scientifically sound in its present form?

Yes

Are the interpretations and conclusions justified by the results?

Yes

Is the language acceptable?

Yes

Do you have any ethical concerns with this paper?

No

Have you any concerns about statistical analyses in this paper?

No

Recommendation?

Accept with minor revision (please list in comments)

Comments to the Author(s)

Review 2- (new title)

Diverse stem-chondrichthyan oral structures and evidence for an independently acquired acanthodid dentition.

Dearden, R. P. & Giles, S.

General comments.

Much improved. The included comments are mostly minor and easily corrected. A useful and now, content-wise, distinct contribution to the study of shark origins, early evolution of jawed vertebrates, and the phylogeny of dental systems.

Detailed comments.

Introduction

L58-61. Clarity: are the dentitions or bones arranged radially? The dermal bones of what taxon are uncertainly homologous to crown gnathostome dental arcades?

L61-66. Another sentence that needs sorting out. Something about the early history of early osteichthyans – too many ‘early’s creates ambiguity – is this about ontogeny?

L85. Teeth not ‘tooth’.

L87. indicates not ‘indicating’

L88. taxa or clades, not ‘animals’

L103. Other not ‘others’

L104. Italicize ‘Gladbachus’.

L271 and elsewhere: check figure numbers in case there’s a general slippage. Here, ‘1f’ should be ‘2f’.

L272. Remove ‘ref’.

L288. 1b should be 2b.

L386. ‘beunmineralized’

L453. Italicize ‘Doliodus’

L458. Decay is a process. The Bayesian analysis is not a process or treatment of the max. parsimony results.

The Bayesian analysis simply failed to find the Acanthodii clade in some or any trees, sufficient to register in the maj. rule summary.

L468. Orvig didn't 'suppose' – he hypothesized, estimated, or conjectured.

L485. I like the notion that toothwhorls are dentigerous jaw bones rolled-up. Very satisfying.

L518. Triangular or conical?

L519. Cryptic homology? Oh dear. Sort this out – what do you mean? Something is homoplastic, but what? And is some other part of this system thought to be homologous?

L531-532. (except for *Acanthodopsis*) – no parenthesis required.

L545-549 – neither of the refs cited is original on this matter (#65 is third-order review/summary). Suggest a ref. from Moya Smith, perhaps?

L552 – more conventionally shark-like taxa?

L572. Un-bold the text.

L576. present, not presents

L582. That that

L587-592. First to third sentences – there's a simple systematic statement buried within but obscured by ...words. Re-think; re-phrase.

L613. Comparative to what? This piece of string is relatively long.

L635. More animals: remove.

L.688+ Tooth shedding scenario – yes and no. Don't forget the presence of generative tooth series in stem chimaeroids such as *Debeerius* and *Helodus*. Are these tooth shedders or retainers?

Extra typos – initiall & chondrichhthyan

L698. Shedding evolved multiple times. Yes. Good point & worth repeating.

L705. Frey et al include a more limited suite of stem chondrichthyans, thus deficient?

L713. A period of inferred or estimated rapid gnathostome evolution.

7L14. 'in of'

L720+. Section 4. Conclusion. Cut? Redundant text unless a summary is required, in which case relabel as 'Summary'.

Decision letter (RSOS-210822.R1)

Dear Dr Giles

On behalf of the Editors, we are pleased to inform you that your Manuscript RSOS-210822.R1 "Diverse stem-chondrichthyan oral structures and evidence for an independently acquired acanthodid dentition" has been accepted for publication in Royal Society Open Science subject to minor revision in accordance with the referees' reports. Please find the referees' comments along with any feedback from the Editors below my signature.

You will see that Reviewer 1, who has seen previous versions of the paper, remains critical, but I and the responsible AE have decided that, given the more positive view of other reviewers (who consider this version of the paper and previous versions) that it is appropriate to publish the paper at this stage and allow the readership to form their own view of it. Please consider whether, in this final revision, you can make further minor changes, e.g. of clarification, that might address some of Reviewer 1's concerns.

Please submit your revised manuscript and required files (see below) no later than 7 days from today's (ie 12-Oct-2021) date. Note: the ScholarOne system will 'lock' if submission of the revision is attempted 7 or more days after the deadline. If you do not think you will be able to meet this deadline please contact the editorial office immediately.

on behalf of Peter Haynes (Subject Editor)
openscience@royalsociety.org

Associate Editor Comments to Author:
Comments to the Author:

Thank you for engaging with the reviewer comments. As you'll see there, remain concerns from one of the reviewers regarding the work; however, given that - in the earlier round of reviews and now in this iteration - a number of reviewers were in favour of publication following revisions, we are going to ask you to conduct a further round of revision before accepting the paper (assuming the editors are satisfied by the changes made, and your rebuttals of critiques). Whether the paper is as problematic as the more critical reviewer suggests, we are of the view that it would be more productive for the paper to be available to the community, who can then engage with the work and - if it appears needed - offer a formal rebuttal: the journal encourages open debate and discussion of the published literature.

Reviewer comments to Author:

Reviewer: 1

Comments to the Author(s)

The most harmful problem of this manuscript is that the authors refuse to present a complete picture of what has been known about early gnathostome dentitions, but only selected information has been shown. For instance, they clearly don't want to take on board the significance of the transverse organisation (file-like arrangement) of the dentitions in stem osteichthyans and acanthothoracid stem gnathostomes, which is shared with chondrichthyan tooth whorls and tooth files, but distinct from the acanthodian dentigerous jaw bones. Consequently, the authors produce a false image that tooth whorls and dentigerous jaw bones can be united as a synapomorphy of chondrichthyans, based on the characters that are in fact general to the gnathostome total group. This will distort the understanding of dental evolution. Although this manuscript focuses on acanthodians, a much more serious (though can be brief) review of various early gnathostome dentitions is expected in the Introduction/Discussion. The authors should not just cite a bunch of papers and then run away, saying that other taxa or oral structures are poorly understood, poorly characterized, etc.

Line 21

"parallel, continuously replacing" is not very informative. It means nothing for people that are not familiar with shark dentitions. I suggest you clarify the transverse organisation of the files and the conveyer-belt replacement.

Line 22

the site-specific replacement?

Line 54

What do you mean by "partially" here?

Line 62-64

The dentitions of stem gnathostomes are definitely relevant to the evolution of teeth, no matter how "strange" they are. They are not poorly characterized, and we do have understood quite a bit.

Line 77

This is not an accurate description of the early osteichthyan dentitions.

Line 90-91

The long history of file-like arrangement can be traced back to the basal jawed stem gnathostomes. The file-like arrangement also occurs in stem osteichthyans, rather than limited to chondrichthyans.

Line 94-95

at the labial/outer jaw margin

Line 98-101

"file-like teeth" and "file-like whorls" sound strange. Does the former mean files of teeth that are not fused into a whorl (non-joined teeth), or does it includes the whorl condition? "File-like teeth" sounds like a description of tooth shape. Whereas all tooth whorls bear tooth files, in other words, there are no "non-file-like whorls". Also, there is an intermediate condition that the teeth are not fused by a common base, but post-functional teeth are retained and packed at the outer margin of the mouth, which looks like a tooth whorl. Shedding or non-shedding is a key difference of the different conditions of file-like organization, and thus should be mentioned in

the description of the different conditions. Yes, it has been mentioned in the following text, but it just makes it more confusing for readers who are not familiar with early chondrichthyan dentitions. So far, not a simple definition of tooth whorl has been given, but readers should not be supposed to know what is a tooth whorl.

Line 106

Not sure which mode of tooth replacement you are talking about. Not understand why the topic of file-like organization suddenly becomes the mode of tooth replacement, and then jumps to tooth whorls in the next sentence. As no osteichthyan is mentioned in this paragraph, there should not be a second mode of tooth replacement, but conveyor-belt style tooth replacement.

Line 120-121

It is also unclear whether tooth whorls are shed from time to time.

Line 132-135

This mode of growth has long been well understood since Ørvig 1973, which is based on several identified taxa. Then Smith 2003. This statement is acceptable for Rücklin et al. 2021 only if say "have been shown by microtomography". Since this is a common mode of tooth addition in acanthodian dentigerous jaw bones, it is not that important whether the specimens are identified or not.

Rephrase "wider construction".

Line 137-138

Strange sentence. Are you trying to say "teeth that are not borne by dentigerous jaw bones"? "not part of ... tooth files" is redundant, when saying "not arranged into files".

Line 228

"gnathal plates" is usually used in placoderms. It is better to stick to "dentigerous jaw bones" for ischnacanthids, or just say "jaw bones" for acanthodians regardless it is dentigerous or not.

Line 264-265

"Elgin" is not a known Middle Devonian locality. In fact, there are no Middle Devonian localities at all around Elgin, only Late Devonian ones such as Scaat Craig and Rosebrae. I wonder where this specimen is from.

Line 368-371

Citation of figures is needed. Why does its position suggest an artefact of growth?

Line 381-383

Shown in any figures?

Line 436-437

Why is it flatter but taller?

Line 463

Is the inner tooth row the same as the lingual tooth row? If so, keep the consistency of the terms. Otherwise, do you mean the inner one of the two rows borne by the medial ridge?

Line 465-468

Label the two additional small cusps in the figures. Do you mean they form an additional tooth row?

Line 550-552

Bad sentence construction.

Line 641-642

Do you mean the radial vasculature? Is the point, which the vasculature outwards from, the ossification centre of both the dermal and endoskeletal bones? Make it clearer with shorter sentences.

Line 662-675

The chondrichthyan dentitions with tooth whorls lining along the length of jaw and whorl-like cheek scales are not site-specific either, as in acanthothoracid stem-gnathostomes and early osteichthyans. All those suggest common patterning mechanisms across early gnathostomes, rather than an apomorphic for chondrichthyans. In other words, many other dental elements across gnathostomes can be regarded as “stretched out” tooth whorls, with new teeth added in various orientations, such as lungfish tooth plates.

It is necessary to define “site-specific” here. Is the site relative to the whole jaw or mouth, or to a dermal plate or an element that unite the teeth? It is especially confusing because you also use “site-specific” to describe the dentitions of bony fishes in the abstract.

You are trying to find a dental character to unite the chondrichthyans. It is nice if there are any. But if not, don't make it up. It is not helpful to understand early gnathostome evolution. As you said “Support values aside from the chondrichthyan total group and crown-group nodes are typically low”, in addition to the situation that the osteichthyan-chondrichthyan node is difficult to place among stem-gnathostomes, it is unwise to put things into boxes and seal the boxes. Early vertebrate phylogeny should be based on morphological observations, instead, your interpretation of the morphology is determined by the phylogeny, which is very unstable. Although all acanthodians are considered stem chondrichthyans at the moment, the diversity of early chondrichthyan dentitions is supposed to cast light on the potential evolutionary relationship between acanthodians and other early gnathostome groups. If acanthodians, at least some of them, were placed in the osteichthyan stem, as in the old times, do you still think that the dentigerous jaw bones are just a modified form of tooth whorls, or more similar to tooth whorls than any other kinds of gnathostome dentitions? In terms of “site-specific”, the linear tooth rows of osteichthyans are more comparable, such as the row of fang pairs, the row of marginal large teeth and the row of accessory small teeth are all site-specifically arranged. Developmentally, it does not make sense for the dentigerous jaw bones, a kind of “antero-posteriorly ‘stretched out’ tooth whorls” in your words, to evolve from the labio-lingual tooth whorls.

Line 735-736

The tooth whorls of iniopterygians is more likely a retained primitive character, because many other stem holocephalans also have tooth whorls along the jaw.

Line 872-879

Do you mean the dentition entirely formed from tooth whorls share a last common ancestor and develop independently in the crownward lineage is one of the scenarios? It is confusing what scenario Frey et al. suggested, according to your statement. As one of the major types of dentitions in Acanthodii, why do you say it “being lost in Acanthodii”?

Line 896-899

It will be nice to indicate the clade only with tooth whorls and the clade with a diverse array of oral structures at the nodes of the trees if they do form different clades.

Figure 1

In (a), it looks like each tooth is a “tooth file”. It is better to box the file of teeth and label the box with a single line. It will be nice to have an inset of the lateral view just like in (b) for comparison.

Abbreviations should be explained in the caption. But it is better not to use abbreviations at all since there is enough space, and the consistency should be kept (“tooth file”, “tooth whorl” etc. are not abbreviated).

Supplementary figure 1

Which taxon does (d) belong to, *Taemasacanthus* or *Atopacanthus*? No need to correspond the taxa with the figures twice in the caption, which actually brings up a conflict. (a) and (d) can be both medial views if they belong to the same taxon. (b) is from the box of (a), rather than the lateral view, so all the following figures are wrong. Same for (f). Check the figure citations in the main text. It is also unclear which figures share the same scale bar. Since all taxa are given the same colour, it is better to divide the figures into sections for each taxon by adding some separating lines and labelling each division with the taxon names.

Can the circular ridge be seen in any of the figures? If so, please label it.

Reviewer: 4

Comments to the Author(s)

Review 2- (new title)

Diverse stem-chondrichthyan oral structures and evidence for an independently acquired acanthodid dentition.

Dearden, R. P. & Giles, S.

General comments.

Much improved. The included comments are mostly minor and easily corrected. A useful and now, content-wise, distinct contribution to the study of shark origins, early evolution of jawed vertebrates, and the phylogeny of dental systems.

Detailed comments.

Introduction

L58-61. Clarity: are the dentitions or bones arranged radially? The dermal bones of what taxon are uncertainly homologous to crown gnathostome dental arcades?

L61-66. Another sentence that needs sorting out. Something about the early history of early osteichthyans – too many ‘early’s creates ambiguity – is this about ontogeny?

L85. Teeth not ‘tooth’.

L87. indicates not ‘indicating’

L88. taxa or clades, not ‘animals’

L103. Other not ‘others’

L104. Italicize ‘Gladbachus’.

L271 and elsewhere: check figure numbers in case there’s a general slippage. Here, ‘1f’ should be ‘2f’.

L272. Remove ‘ref’.

L288. 1b should be 2b.

L386. 'beunmineralized'

L453. Italicize 'Doliodus'

L458. Decay is a process. The Bayesian analysis is not a process or treatment of the max. parsimony results.

The Bayesian analysis simply failed to find the Acanthodii clade in some or any trees, sufficient to register in the maj. rule summary.

L468. Orvig didn't 'suppose' - he hypothesized, estimated, or conjectured.

L485. I like the notion that toothwhorls are dentigerous jaw bones rolled-up. Very satisfying.

L518. Triangular or conical?

L519. Cryptic homology? Oh dear. Sort this out - what do you mean? Something is homoplastic, but what? And is some other part of this system thought to be homologous?

L531-532. (except for Acanthodopsis) - no parenthesis required.

L545-549 - neither of the refs cited is original on this matter (#65 is third-order review/summary). Suggest a ref. from Moya Smith, perhaps?

L552 - more conventionally shark-like taxa?

L572. Un-bold the text.

L576. present, not presents

L582. That that

L587-592. First to third sentences - there's a simple systematic statement buried within but obscured by ...words. Re-think; re-phrase.

L613. Comparative to what? This piece of string is relatively long.

L635. More animals: remove.

L.688+ Tooth shedding scenario - yes and no. Don't forget the presence of generative tooth series in stem chimaeroids such as Debeerius and Helodus. Are these tooth shedders or retainers?

Extra typos - initiall & chondrichthyan

L698. Shedding evolved multiple times. Yes. Good point & worth repeating.

L705. Frey et al include a more limited suite of stem chondrichthyans, thus deficient?

L713. A period of inferred or estimated rapid gnathostome evolution.

7L14. 'in of'

L720+. Section 4. Conclusion. Cut? Redundant text unless a summary is required, in which case relabel as 'Summary'.

===PREPARING YOUR MANUSCRIPT===

===PREPARING YOUR REVISION IN SCHOLARONE===

Author's Response to Decision Letter for (RSOS-210822.R1)

See Appendix B.

Decision letter (RSOS-210822.R2)

Dear Dr Giles,

I am pleased to inform you that your manuscript entitled "Diverse stem-chondrichthyan oral structures and evidence for an independently acquired acanthodid dentition" is now accepted for publication in Royal Society Open Science.

The proof of your paper will be available for review using the Royal Society online proofing system and you will receive details of how to access this in the near future from our production office (mailto:openscience_proofs@royalsociety.org). We aim to maintain rapid times to publication after acceptance of your manuscript and we would ask you to please contact both the production office and editorial office if you are likely to be away from e-mail contact to minimise delays to publication. If you are going to be away, please nominate a co-author (if available) to manage the proofing process, and ensure they are copied into your email to the journal.

on behalf of Professor Peter Haynes (Subject Editor)
openscience@royalsociety.org

Appendix A

Associate Editor Comments to Author:

An unusually large number of reviewers have been kind enough to return reports on your paper. While the overall picture appears to be positive, there are enough comments, queries and suggestions that we would like you to revise the manuscript as far as possible to take these concerns into consideration. Good luck!

Many thanks to the editor and to all reviewers for their thorough and thoughtful commentary on our manuscript. We have tried to address all comments with changes, and where that was not possible have given justifications. In the accompanying manuscript file we have tracked changes to make it clear what is new text. The order of some parts of the discussion has been rearranged (see next paragraph): where text has just been moved this has not been tracked to make it clear what text is actually new.

We have tried to chart a course that takes into account the different reviewers' suggestions. On the advice of reviewer 4 we have reworked the introduction and discussion of the paper to focus it more on chondrichthyans. As part of this we have written a more extensive introduction to contextualise the esoteric world of stem-chondrichthyan teeth as suggested by reviewer 3, and have rearranged our discussion to give higher billing to our new results as suggested by reviewer 4. By placing the focus firmly on chondrichthyans we also address several of the points raised by reviewer 1 regarding our treatment of non-chondrichthyans, and hope that it makes the novel aspects of our study clearer.

Reviewer comments to Author:

Reviewer: 1

Comments to the Author(s)

Dearden and Giles described the jaw bones and teeth of five acanthodian genera. Then they summarized different oral structures of acanthodians, and mapped these jaw and teeth characters in the recent phylogenetic analyses. The description was based on uCT scans that do not have an adequate resolution for detailed histological studies. The data was presented by surface rendering and virtual thin sections, but no segmentation of internal structures and no 3D insight. Tissues cannot be identified with certainty and the development of the tooth rows cannot be properly comprehended. This makes it especially tricky to interpret the material of *Acanthodopsis*, which is a particularly important taxon in acanthodian dental evolution. This study provides some new anatomical information. However, since articulated specimens of acanthodians are very rare and histology becomes the key to chondrichthyan evolution, the conclusion that can be drawn from this study is limited.

Comments on materials and methods:

The voxel size of the CT scans at 17.3 μm , 19.5 μm , 22.6 μm , 24.6 μm and 44.9 μm is far too large and can provide information no better than the anatomical level. However, to give interpretations at the histological level, which is necessary for a proper understanding and has been tried by the authors, usually requires high-resolution synchrotron scanning with a voxel size lower than 1 μm .

Reviewer 1 is correct that higher resolution scanning would deliver more detailed histological information but we disagree with the judgement that our CT scans are not suitable for interpreting histology and provide no important new information. Synchrotron scanning is not

feasible for this study, which was an unfunded research project. Access to synchrotron scanning is extremely limited compared to CT scanning and is not easily accessible to all research groups. Even if we were able to bid for time, this process would take the best part of a year, plus, assuming the bid was successful, months of segmenting time. The only synchrotron in the UK is not suitable for this kind of work (SG carried out scanning there in 2019) and travel to synchrotron facilities outside of the UK is not compatible with our current funding options and caring responsibilities. Furthermore, this research is being carried out during the Covid-19 pandemic where there is extremely limited access to museum specimens and travel: the NHMUK has been closed to all but essential time-limited funded project visits, and loaning of specimens to take them for additional scanning is impossible. None of the conclusions of our paper rest upon the detailed histological characterisations that synchrotron scanning would be required to achieve: our scan sets are still sufficiently detailed to show the pattern of tooth addition in the ischnacanthid jaw bones as well as their overall structure, and the distinct tissues in *Acanthodopsis* and *Acanthodes*. Even at the anatomical level our study provides valuable new comparative information that will help others interpret fossils: the first detailed characterisation of a mandibular splint and its relationship to the Meckelian cartilage, the first detailed characterisation of a range of ischnacanthid jaw bones (Rücklin *et al* 2021 focussed only on the tooth rows), and the first detailed characterisation of an *Acanthodopsis* jaw bone. Synchrotron scanning of these specimens may be an avenue for future studies to explore, but is far beyond the scope of our project.

One of the acanthodian genera, *Ischnacanthus*, has been studied by Rücklin *et al.* 2021, which also using microtomography but with a much higher resolution (synchrotron scan with voxel size 0.74 μm). The authors stated that the teeth lack enameloid, contra the result of Rücklin *et al.* 2021, which may or may not be true. However, considering the great difference in the resolution of the scans between the two studies, this statement is not convincing: enameloid is unlikely to be visible in the lower-resolution scans of this study. Compared to the classic study of Ørvig 1973, this study is not able to enhance the understanding of acanthodian dentitions to a great degree.

Careful reading of Rücklin *et al.* shows that they do not actually characterise the histology of *Ischnacanthus* at 0.74 μm but rather the histology of an ischnacanthid jaw bone of undetermined taxonomy. Morphologically this bone is more similar to the *Taemasacanthus* jawbone we describe than it is to *Ischnacanthus*. Rücklin *et al.* scan and figure the outside of an *Ischnacanthus* jaw bone (their fig. 1a), but all histological data comes from the ischnacanthid jaw bone because their *Ischnacanthus* data has insufficient contrast to be able to examine the histology (as can be seen in the tomographic dataset they provide in the supplement).

Reviewer 1 is correct that our scan data is significantly lower than that of Rücklin *et al.* However, the layer of enameloid in the ischnacanthid described by Rücklin *et al* is reported as being 5-70 μm wide. Our reported voxel size for the ischnacanthid taxa in our sample is in the region of 20 microns, so we would expect that the thicker portions of any enameloid layer, if present, would be visible in our scans. We acknowledge that a thinner layer may not be resolved in our scans even if present, and have reworded these parts of the text to make the limitations of our methodology clear.

Comments on references and the conclusive figures:

Many contra papers have pointed out that what Rücklin & Donoghue 2015 [ref. 4] described is not a tooth-bearing bone and it is impossible to determine whether it belongs to

Romundina. Rücklin himself has admitted his mistake in public. However, this manuscript, which does not provide any supporting evidence for it, cited Rücklin & Donoghue 2015 without criticism. This is coupled with the coding of *Romundina* (with teeth) in the matrix, implying the authors still believe Rücklin & Donoghue 2015. Whereas CPW.9, which does certainly have teeth, is definitely NOT *Romundina*. Thus the tree showing the distribution of teeth is misleading.

We agree with the reviewer that the element in Rücklin and Donoghue 2015 is not a tooth-bearing bone and its affinity to *Romundina* is unclear. Our reference to this paper is in the context of a brief review of published stem-gnathostome dentitions because it also contains relevant data pertaining to the supragnathal of *Compagapiscis*; we do not refer to the '*Romundina*' specimen at all.

The reviewer is correct that we made an error in the figure with regard to *Romundina*, although we note that it was coded correctly in the underlying phylogenetic analysis (i.e. with the presence of teeth unknown). We have updated the figure to include a taxon for which there is no associated ambiguity (*Austroptyctodus*).

In the tree of figure 6, stem gnathostomes are only represented by three taxa; actinopterygians and sarcopterygians only have a solo representative, respectively; no stem osteichthyans at all. The “placoderm-osteichthyan transition” is thus not reflected. This has oversimplified the dental evolution of gnathostomes, even though this manuscript focuses on acanthodians. As a consequence, the picture of vertebrate tooth evolution is distorted. Key taxa are not shown, and discussion on the dental diversity of early gnathostomes and their potential evolutionary relationship with the acanthodian dentitions has been dodged. A large number of recent discoveries of stem gnathostomes and stem osteichthyan dentitions are neglected, but only cited as “The advent of micro-computed tomography has led to a renewed interest in tooth evolution and development in Palaeozoic gnathostomes. These have mostly focussed on stem-gnathostome ‘placoderms’ [2,5,6,8] and osteichthyans [9–14].” No comparison of morphological and histological details, and no discrimination between supported and falsified theories among previous studies, and thus little meaningful contribution to the literature.

As the reviewer notes, this manuscript focuses on dentitions in stem chondrichthyans, all the more so now that we have reframed the paper in line with Reviewer 4’s suggestions, and we explicitly do not seek to contribute to the ongoing debate around the ‘placoderm-osteichthyan transition’. For this reason, we have chosen a recent phylogenetic analysis targeting the chondrichthyan stem-group, with relatively few stem-gnathostomes. The tree aims to survey stem-chondrichthyan dentitions, not the entire gnathostome total-group.

Although a survey of the “placoderm-osteichthyan transition” is beyond the purview of our study we do believe that the data our manuscript will be very relevant for other workers addressing those questions. As the reviewer thinks it would add to the manuscript, we have included a conclusions paragraph suggesting some of these broader implications. We have also expanded our survey slightly in the introduction to try and better capture the range of morphologies seen in placoderms and osteichthyans.

A detailed discussion should be made for each icon in figures 6 and 7. Does “x” indicate absent? It should be denoted in the figure legend. What does “teeth ankylosed” mean? The teeth of *Psarolepis* and *Cheirolepis* are definitely ankylosed in the dermal jawbones. Why they are marked as “x”? A tooth whorl is formed by tooth based joined, but each tooth whorl

is a separated dermal element, how can the whorl bases joined? What does “whorl bases joined” mean? It does not correspond to any oral structures that are mentioned in Discussion.

The terminology in Figures 6 and 7 relates to the characters underpinning these morphologies in the data matrix. Characters such as ‘teeth ankylosed’ are well established in early gnathostome data matrices. We have edited the figure, legend and text to include specific callouts to related character numbers in the data matrix and standardised language throughout. We have also added a line to explain that ‘x’ indicates absence.

The original descriptions that the distribution of the colored icons based on should be cited for each taxon in the figure legend.

The distribution of the coloured icons is based on codings made in the phylogenetic analysis, and we have clarified this in the legend.

Minor comments:

Line 43-44

“lateral to and overlying endoskeletal jaw cartilages” would better correspond to “outer and inner dental arcade”, rather than “inner and outer”.

We have amended the text accordingly.

As in many places throughout the text, “endoskeletal” can be deleted, since there is no exoskeletal cartilages.

We have surveyed uses of endoskeletal and have removed them where redundant. We have retained its first use: while it is technically redundant, we feel that use here should help readers unfamiliar with these structures follow the text (as suggested by reviewer 3).

Line 46

“labially-directed” should be “labiolingually-directed”.

We have amended the text accordingly.

Line 55-57

Ref. 18 is about the oral scales of jawless fish thelodont, which cannot be assigned to the chondrichthyan total group.

In addition to thelodont oral scales, the authors of ref. 18 synchrotron scanned and figured an ischnacanthid acanthodian tooth whorl for the first time, hence our reference to it.

Line 57-60

This long sentence needs to be rephrased. “a staggering array” can be used to described tooth organization, but it seems not the case here and is thus confusing.

As also pointed out by reviewer 4 “a staggering array” is hyperbolic: we have changed the wording.

What is the logic to lump the dermal oral structures together with commas? Teeth on dermal plates (do they mean the dentigerous jaw bones?) or tooth whorls and extramandibular dentitions regard the location and tooth-bearing bones, whereas gracile and molariform regard the tooth shape. References should be cited for the specific dermal oral structure they

apply to, for example, ref. 15 should be cited behind “tooth whorls” and “gracile”, ref. 20 should be cited behind “tooth whorls” and “molariform teeth”, etc.

This has been addressed by the rewritten longer-form introduction also requested by reviewer 3

Line 188

Unlike those in the bone, the vascular canals at the tooth bases are not parallel. But to determine whether they are randomly oriented, segmentation and 3D visualization is required. Can they be considered as pulp canals?

We have added a supplementary figure showing orientation of vascular canals throughout the specimen for both *Taemasacanthus* and *Atopacanthus*. The vascular canals are distinct and separate from the pulp canals—both are visible in *Atopacanthus*. We have added more description of the vascularisation to the text.

Line 189-191

Do the cusps of the medial row also show a size gradient and overlap between cusps? Is the vascularization related to their size?

The medial cusps are not positioned closely enough to overlap, and vascularisation does not appear related to size. We have amended the text to include this.

Line 192

The crown of all teeth in figure 1f looks vascular too. If the crown has fewer vascular canals, is it because they have been infilled?

This is a good point: we have incorporated this into the text.

Line 195-196

In figure 1d, it looks like the tenth cusp is not only smaller than the ninth cusp, but also the seventh cusp.

This is true: we have highlighted this in the text.

Line 198

Is the bone around the eighth cusp also damaged?

Not as far as we can tell – amended the text to clearly state this.

Line 199

Tomogram sections cutting through the ninth cusp and the neighboring cusps are required.

We have added this to figure 1.

Line 200

Are the vascular canals of the ninth cusp closely connected to those of the eighth and tenth cusps, and the bone?

No – we have made this clear in the text.

Line 201

Does the medial position of the ninth cusp correlate to the most damaged part of the eighth cusp?

It is not possible to tell as the entirety of the eighth cusp is damaged.

Line 206

It seems the ridges are all conspicuously tuberculated in the cusps of the anterior half of the tooth row. Just the number of tubercles depends on the length of the ridges.

This is accurate: we have amended the text to make this clearer.

Line 207

Does it mean they cannot occlude anteriorly?

As the upper jaw bone of *Taemasacanthus* is unknown, we cannot say for sure. However, this lingual ridge is present in other ischnacanthiforms, and doesn't seem to prevent their teeth occluding (Burrow, 2004¹)

Line 244

What is the histological relationship between the two additional cusps and the lingual rows of teeth? Are the two additional cusps similar to those on the medial ridge?

These cusps are ostensibly similar to those of the medial ridge in that they are of similar size and vascularised similarly. They seem to be independent from the lingual teeth, rather than lying on the dermal bone plate. We have amended the text to make this clearer.

Line 250

Again, the cap looks vascular too. The basal part of the pulp cavity of the younger cusps is more widely open, but vascular canals clearly extend to the crown. The pulp cavities of the older cusps are less widely open, demonstrating infilling is ongoing from the crown to the base as more dentine is deposited centripetally.

This is a good observation which we have incorporated into the main text.

Line 305-306

It is very important to show a section through the largest tooth and its anterior and posterior neighbors.

As stated above, we have added sagittal sections through the lingual tooth row to Fig 1. A section through the largest tooth of the lateral row is seen in Figure 1g.

Line 421-423

According to the phylogenetic result, it is impossible to tell whether the taxa with tooth

¹ Burrow (2004) Acanthodian fishes with dentigerous jaw bones: the Ischnacanthiformes and *Acanthodopsis*. *Fossils and Strata* (50)

whorls comprising the entire dentition is more crownward than those with tooth whorls limited to the symphysis.

This is not the case: in our tree we resolve a clade (*V. waynensis*->*Hamiltonichthys*, which is inclusive of the crown node) that includes all taxa with a dentition comprising only tooth whorls. All other taxa in that clade either do not have tooth whorls, or the condition is unknown.

Line 482

How to define dermal mouth plates? If it means large dermal jaw bones, it is, of course, only absent in the chondrichthyans without dentigerous jaw bones.

We have rewritten this section to make this clearer.

Line 486-487

Does “occlusal plates” correspond to the “dermal plate smooth” in the figures? If so, they should be referred to in the same way.

We have amended the figure to make this clearer.

Line 489-491

Not all the references cited here have ever mentioned occlusal plates or mandibular splint. It is necessary to denote what the occlusal plates were called in the original description of these taxa. It will be better to make a drawing of a generalized acanthodian head above figure 6 to explain the difference between occlusal plates and mandibular splint by showing their location in the mouth. Otherwise, it is confusing, because none of the taxa described in this manuscript has occlusal plates.

We have included an occlusal plate in the summary figure requested by several reviewers to make this clear. We have also clarified the range of terminologies used.

Line 499

Should “already” be “almost”?

We have addressed this ambiguity by removing the word, which was unnecessary.

Line 555-557

“a single direction in isolated files on an underlying dermal plate” is a common feature of vertebrate dentitions and one of the basic criteria to recognized true teeth. It by no means supports the comparability between dentigerous jaw bones and tooth whorls.

We agree that this is a character broadly applicable to teeth. However, whether this is a plesiomorphic character or a synapomorphy this comparison of tooth whorls and dentigerous jaw bones still stands.

Line 561-565

What does “isolated files” mean? If “isolated” means each file grows on a separated bone, or if “files” indicates a labiolingual organization of teeth, then the dentigerous jaw bones, which bear multiple anteroposterior tooth rows, do not have isolated files. It is impossible to unite the dental development of dentigerous jaw bones and tooth whorls in this way. It is not uncommon for a tooth whorl to bear three files. Multiple non-shedding tooth files have been seen in acanthothoracid marginal jawbones, and the inter-file spacing is very clear too if that is what “isolated” means. Multiple shedding tooth files are also found in stem osteichthyan

marginal jaw bones and tooth cushions, as well as sarcopterygian parasymphysial tooth whorls. Sauropods could generate teeth so quickly that their tooth families appear as the tooth files of modern sharks, while the tooth families of hadrosaurid dental battery are comparable with the chondrichthyan tooth whorls or files of tooth retention. Therefore the lingual tooth addition represented by tooth whorls is likely a primitive setting that is shared by the basal members of different groups of gnathostomes, and with modification, it has been carried on by osteichthyans. The anterior tooth addition is unique to the acanthodian dentigerous jaw bones. Nevertheless, the tooth rows on the dentigerous jaw bones are more comparable to other dentitions with a radial organization, such as in arthrodiras and lungfish. All these growth patterns are related to the developmental relationship between odontodes and the underlying bones, and thus similar patterns can appear again and again. Hence, it is more important to explore the developmental mechanism than only extract phylogenetic signals. All in all, isolated files can never be a character uniting chondrichthyans.

We have rewritten this part to try and more clearly define what we mean by isolated file. We agree with the reviewer 1 is right that there is evidence that lingual tooth addition may be plesiomorphic for gnathostomes, although the stem-gnathostomes this is based on are not firmly phylogenetically placed and sit amidst an unsettled set of taxa with different tooth morphologies, while the stem-osteichthyans comprise only isolated jaw bones. In none of these taxa are the teeth arranged into clearly separated files, and files they are not located on site-specific locations on the dermal bone as they are in e.g. ischnacanthid dentigerous jaw bones. Our text now makes a clearer comparison between these stem-gnathostome and early osteichthyan structures and chondrichthyan dentitions.

Whether or not it is a plesiomorphic character this comparison of tooth whorls and dentigerous jaw bones still stands.

Line 592-593

Dermal jaw bones are not likely to present in taxa least proximate to the chondrichthyan crown either.

This sentence is no longer in the manuscript due to the rewrite suggested by reviewer 4

Line 595-597

There is not enough evidence of the first step that a dentition with both dentigerous jaw bones and tooth whorls shifts to that with tooth whorls only. In the phylogenetic analysis, dentigerous jaw bones only occur in a clade of acanthodians, which may indicate specialization. For the second step, please read Johanson et al. 2020 Integr. Comp.

We weigh up different phylogenetic arrangements in this manuscript (see figures 6 and 7) and openly consider this possibility.

Line 599

Frey et al. 2020, as the tree in figure 7d, does support a reverse shift that tooth whorls occupy the full length of the jaw in basal acanthodians, and are then partially replaced by dentigerous jaw bones in the clade of ischnacanthids.

We acknowledged this, both in the original manuscript and in the rewritten version.

Line 603-609

Here the discussion suddenly shifts to the evolutionary relationship with osteichthyan jaw bones and teeth, and the conclusion seems to be that the dental system of acanthodians and osteichthyans evolve completely independently. However, the authors ignored the fact that the teeth of stem osteichthyans are commonly organized in the same way as the chondrichthyan dentitions entirely made up of tooth whorls, just the whorl bases joined into a large dermal bone. “Whorl bases joined” is exactly what the dark blue irons in figures 6 and 7 indicate. This may be convergent but is well worth mentioning.

Due to the rewrite this comment is no longer specifically relevant, but it is still a useful point about osteichthyan tooth whorls which we have now mentioned clearly in the revised manuscript.

Line 613-623

That is right. Therefore, the authors should focus on the actual comparative morphologies, respect any conflicting morphological data, rather than be eager to make up hypotheses by playing with the trees.

We are unclear what the actionable criticism is in this comment. We initially focus on comparative morphologies and then explicitly consider alternative phylogenetic hypotheses: we are unsure what is meant by ‘playing with the trees’. All of the trees figured in our paper result from phylogenetic analyses targeting the chondrichthyan stem-group, and we are up front about the fact that they differ in topology (that’s why we figure so many).

Line 617

The dentitions described in ref. 6 are not at all difficult to interpret. The histology is perfectly preserved in 3D and more fully documented than the acanthodian material in this manuscript. They are just difficult to fit in the conventional/the authors’ own hypothesis. But just because of that, these dentitions should not be dismissed to avoid addressing the challenges posed by the authors. A detailed comparison with all these dentitions is supposed to be made in the Discussion.

We did not mean to imply that the dentitions in this reference are difficult to interpret or does not fit with any particular hypothesis, rather that the morphologies are difficult to compare to those of extant taxa. They are also difficult to interpret in a phylogenetic context as all four of the taxa described in Ref 6 are recovered in a polytomy with *Romundina*, and it is widely acknowledged that stem-gnathostome relationships are far from settled.

Regarding the charge that we are dismissive of these stem-gnathostome morphologies, this also was not our intention. Our study is heavily focused on stem-group chondrichthyans, more so following the reviewer 4 mandated rewrite, and it is difficult to summarise the panoply of total-group gnathostome dental morphologies. Ref 6, after all, does not mention dentigerous jaw bones once. We have tried to link our study into this wider field in the rewritten conclusion however, and we very much hope that our data will be used by workers looking at gnathostome dentitions more broadly.

Reviewer: 2

Comments to the Author(s)

The authors provided a great study and review of oral structures, especially teeth in early chondrichthyans. Great data set, neat method, nice figures, and appropriate analyses.

I feel that the paper would benefit from either more taxa labels in figure 6 or clear definitions of group memberships. Given the instability of some taxa and groups it might be easier for the reader if groups are made more clear.

We have added taxon labels for all clades that we can be confident of monophyly for, namely osteichthyans, the gnathostome crown group, and the chondrichthyan total and crown groups. It is well established (and explicitly stated in the text) that ‘acanthodians’ are a paraphyletic array on the chondrichthyan stem. We do not label purported clades such as acanthodiforms, ischnacanthids and diplacanthids in figure 6 because we do not recover support for their monophyly. We feel that labelling all taxa with the groups that they have previously been affiliated with would make the figure too complex and messy. Instead, we discuss the non-monophyly of these in the phylogenetic results paper. During the review of oral structures, we refer to putative acanthodian groups giving examples of genera that are in our phylogenetic figure. We hope that these results will contribute to a more robust hypothesis of stem chondrichthyan relationships in the future.

Reviewer: 3

Comments to the Author(s)

I really enjoyed the CT scans and anatomical discussion of teeth and teeth-like structures presented in your manuscript. The discussion is well-written and explains the different types of teeth structures very well and for the first time clearly illustrates the evolution of teeth, teeth-like structures, and tooth-bearing bones in these groups. My main issues lie in the introduction which is incredibly short. Please find detailed comments in the attached document ("**RSOS-210822 review-submit.pdf**").

I enjoyed the discussion

Other suggestions:

Add posterior probabilities to figure 6

We have added posterior probabilities to this figure.

Line 534-537 -> but with low support. Which seems common for the data set... is this ok? Support values across early gnathostome phylogenies are notoriously low. While this is not ideal, it is to be expected for the taxon set and we are not unduly concerned by it.

Introduction: After reading through the whole document the intro is lacking important information to set up the rest of your story. Paragraph 1 begins to outline the evolution of teeth in these groups and then paragraph 2 jumps right into why CT has increased our understanding of the evolution of teeth and jaws. I suggest adding a paragraph in between that begins to outline some of the distinctions in tooth morphology and attachment seen across the evolution of regenerative dentitions. This would help greatly when it comes to your morphological descriptions which is in large the main results of this manuscript. An additional paragraph between 2-3 is also needed to better highlight why the

placements of these groups have been particularly problematic and why dental characteristics may be misleading. I realize that RSOS has page and word restrictions however I suggest comments below to help cut length from the anatomical description so that a more appropriate introduction can be made. By spending time in the introduction defining anatomies, you can also eliminate those descriptions throughout the materials and methods.

As suggested by Reviewer 3 we have expanded the introduction by adding two paragraphs, and it now covers all of the suggested points. We have also tried to use this to address some of the comments of reviewers 1 and 4.

Methods: CT scanning: these images and scans are great; It would be beneficial to have a table that IDs the specimen, museum number, and scanning setting.

We have added a table to the supplement detailing this, and have removed scan parameters except voxel sizes from the main text to economise space.

Results: Overall, this anatomy gets confusing, I think it is because there is no clear description of these bones earlier and no analogy to what is observed in extant taxa. There are several instances throughout the results where you begin to talk about information or comparisons that belong in the discussion. Removing these will help to shorten this section. Additionally, a well-placed diagram of this anatomy will help cut down on some initial explanations of where these bones generally are.

We hope we have addressed these comments by expanding the introduction and incorporating a summary diagram (as also suggested by reviewers 1 and 4). These structures generally do not exist in extant taxa so analogy is difficult. We accept the reviewer's point that we justify some of our interpretations of anatomy with reference to the literature in the results section. This is common in palaeontological literature, and we believe makes the results easier to follow, in particular now that we have made the above amendments.

Line 167: suggest for labial cartilage attachment – this is discussion and not results and should be removed It would be immensely helpful to have a general schematic of this anatomy to reconcile the CT scan morphology

See note on interpretive results vs discussion above. We have revised the text to suggest that this feature is more likely for a ligament attachment. We have also included a summary figure showing general anatomy in an ischnacanthid jaw bone.

Line 176-177: like that observed in thin sections of other ischnacanthid dentigerous jaw bones--- this is discussion and should be removed from the results You mention histological results but do not present any histological methods. What histological preparations were done? If this is information solely produced from other manuscripts it belongs in the discussion otherwise, please add a section regarding the histological preparations, sectioning, and staining.

In line with other palaeontological literature, we refer to histology interpreted from CT scan data (in keeping with the definition of histology as “the microscopic study of tissues”), rather than ‘classic’ histological stained sections.

Line 193: starting at “both grow” – do you mean that new teeth are being added at the front of the jaw? The phrasing “addition of new teeth anteriorly” is confusing, I would consider rephrasing to be more direct

We have rephrased this to make it clearer

Line 196: early in the results you said biting edge and now you are saying lingual, what is the difference between these two places? Maybe you could explain this earlier or again add it as a small schematic.

We have rephrased this to make it clearer: the biting edge comprises the portions of the jaw at the contact points of the teeth, and is therefore more expansive than the lingual edge (which only refers to the lingual row of teeth).

Line 212-213: this is discussion and should be removed

See note on interpretive results vs discussion above

Line 214-19: this is discussion and should be removed. The only part that results is “The tissue forming this is notably ill-formed, and appears to lack a solid perichondral covering”

See note on interpretive results vs discussion above

Line 227: again, you mention histology but there is no mention of this in materials

See comments about histology above

Line 320: starting at “Burrow” is discussion

See note on interpretive results vs discussion above

Line 329: I think you can remove the part saying that the articular was already described since you do not go into any detail about it here

We put this in to refer the reader to a detailed description in case they required it so prefer to leave it in.

Discussion The section on distribution of oral structure is great! I really appreciate how you have broken down the paragraph by the type of tooth structures seen in the fossil record and how that morphology is reflected in different groups

Line 418: do the tooth whorls of stem holocephalans look/ form the same way as in stem Chondrichthyes?

The whorls we refer to are quite poorly characterised, so our knowledge is limited. They look superficially similar to those of stem-chondrichthyans but are probably not homologous due to their phylogenetic distance. We have added more discussion of this to the text.

Line 420: describing the morphological disparity of tooth whorls here with distribution is confusing because it sounds like you are referencing a phylogenetic distribution. I would suggest rephrasing. Try some like “there is significant variation in the positioning of tooth whorls across groups such that....”

We have modified the text to try and make this clearer.

Line 424: shed and resorption contradict each other. You can just say that Steichen tooth whorls are reabsorbed

The osteichthyan teeth in these cases are shed by resorption of the tooth base tissue leading to the crown being shed. However this was obviously poorly phrased: we have rewritten accordingly to make it clearer.

Line 443-446: I think it would be worth while here to go into detail about why this result is so atypical and why despite your findings and dense review you hypothesize that tooth files are homoplasious.

We have rewritten this section to clarify our meaning. The taxa that display the condition are scattered across the tree, indicating that the condition is homoplasious.

Line 453: it would be worth it to expand on this last sentence and place it in the broader literature looking at the evolution of dentitions. As it stands the sentence does not quite make sense but is an important part of the story and should be given more in-depth presentation.

We have amended the text of this sentence to try and make this clearer. This subject is also partly discussed in what is now the first part of the discussion.

Line 467: again, the last sentence of this paragraph is immensely important, but you provide no explanation for how it fits into the rest of the ideas and information you are presenting. Please expand on why we expect this trait to not carry any signal

We have rewritten this final sentence to address this.

It would be beneficial in the discussion to suggests alternative means for reconciling these difficult phylogenies.

This is now addressed in the concluding paragraph of the discussion.

In addition to discussing the pitfalls of the ancestral state reconstruction you could mention some of the functional hypotheses and methods used to tease out dental characteristic. Such methodologies can be found in:

- D'Amore, D. C. (2015). Illustrating ontogenetic change in the dentition of the Nile monitor lizard, *Varanus niloticus*: A case study in the application of geometric morphometric methods for the quantification of shape–size heterodonty. *Journal of Anatomy*, 226(5), 403–419. <https://doi.org/10.1111/joa.12293>
- Hulsey, C. D., Cohen, K. E., Johanson, Z., Karagic, N., Meyer, A., Miller, C. T., Sadier, A., Summers, A. P., & Fraser, G. J. (2020). Grand Challenges in Comparative Tooth Biology. *Integrative and Comparative Biology*, icaa038. <https://doi.org/10.1093/icb/icaa038>
- Cohen, K. E., Weller, H. I., Westneat, M. W., & Summers, A. P. (2020). The Evolutionary Continuum of Functional Homodonty to Heterodonty in the Dentition of Halichoeres Wrasses. *Integrative and Comparative Biology*, icaa137. <https://doi.org/10.1093/icb/icaa137>
- Mihalitsis, M., & Bellwood, D. (2019). Functional implications of dentition-based morphotypes in piscivorous fishes. *Royal Society Open Science*, 6(9), 190040. <https://doi.org/10.1098/rsos.190040> Considering the possible function of teeth and teeth like structures may help in recovering how similarly assembled dentitions evolved.

We thank the reviewer for their suggestion but we have not added these references as we feel they would be more appropriate in a paper that aims to test explicit functional hypotheses, which we do not attempt.

This discussion clearly (I think for the first time) assembled a broad fossil record of dental morphologies. What is lacking is a deeper explanation of how these results relate to greater evolutionary hypotheses about the evolution of dentitions (examples are inside-out vs outside in, etc.). I suggest that the authors spend some time more clearly identifying how these differing morphologies clarify our understanding of the overall evolution of teeth and include alternative testing methods to recover a stronger phylogenetic signal.

We have expanded our discussion to incorporate some of this. However, we deliberately do not attempt to tackle the ‘big questions’ of tooth evolution because we focus on chondrichthyan teeth, and early chondrichthyan (and gnathostome) relationships are not robust enough to answer these questions at this point.

Figures: All CT scans need scale bars

We have included a scale bar for each figure panel.

When are you referring to histological sections in the results are you meaning the tomographic sections? At the resolution of your scans there is going to be a difference between true histological sections and virtual ones. If there is no traditional histology presented in this paper the text should be adjusted to reflect the actual methodology.

As discussed above, it is common in the palaeontological literature to refer to histology from tomographic sections.

Reviewer: 4

Comments to the Author(s)

General comments

This is a valuable review and analysis of early chondrichthyan ‘acanthodian’ dentitions, with emphasis placed squarely on Acanthodopsis and ischnacanthids.

As a result, the present article needs to distinguish itself clearly from Rucklin et al. 2021 “Acanthodian dental development and the origin of gnathostome dentitions” (nat. ecol. & evol. Reference #7).

Both works focus on ischnacanthid jaw bones and dentitions; both use computed tomography to investigate hard tissue histology and infer patterns of growth; both present hypotheses concerning the evolution of dentitions in early jawed vertebrates from a stem-chondrichthyan perspective. However, Rucklin and colleagues are concerned with teeth in the last common ancestor of crown gnathostomes whereas the present work concentrates on dental diversity in stem-chondrichthyans. The takeaway message from the present work is that despite noise in the data set, Acanthodopsis is confirmed as an acanthodid (not an ischnacanthid), and, as such, evolved its teeth independently within an otherwise toothless clade. Interesting!

For these reasons, I recommend retaining the guts of the paper - data, descriptions, analyses

and results, but shift the emphasis in the discussion and intro.; changing the title and abstract accordingly.

We agree with the reviewer and have revised the text accordingly. We have rewritten the introduction to incorporate these suggestions and extend it as advised by Reviewer 3. The Discussion has been rearranged to focus more clearly on our new results and on chondrichthyans. The abstract has been rewritten to reflect this shift in focus. We have also updated our title to: “Diverse stem-chondrichthyan oral structures and evidence for an independently acquired acanthodid dentition”

Clearer images of the CT histology would help – perhaps line-drawn diagrams depicting where the authors identify boundaries between tissues?

We have tried to rewrite the text to make histological descriptions clearer. Many of the tissue boundaries (especially in *Acanthodopsis*) are challenging to figure from individual tomograms and is clearer looking through the CT slices, which are provided in the supplement. We appreciate the data are not as clear as would be ideal and have expressed caution where necessary in our interpretations.

Might the addition of a comparative figure aid the descriptions and discussion?

We have added this figure as requested by reviewers 3 and 4.

Detailed comments

Abstract

Re-draft with greater emphasis on new findings and how these change or challenge current views.

We have redrafted the abstract and introduction, as outlined above.

L. 56-57 – list the three (?) stem-chondrichthyans described using CT: *Doliodus*, *Acanthodes*, *Pucapampella*, ischnacanthids (various), *Gladbachus*...

Stem chondrichthyan dentitions are varied, but ‘staggering’ (as a superlative) is a bit strong compared with, for example, mammal teeth running the gamut from narwhales to elephants, rodents, armadillos, crab eating seals, and the products of US dentistry.

This is a reasonable point which was also brought up by reviewer 1, and we have amended it in the text.

L. 81 – every hypothesis deserves a reference: Long '86?

We have added this reference in as requested.

L. 88-92 – to be clear, *Atopacanthus* is known only from disarticulated remains?

Yes, although a scapulocoracoid and pectoral fin spine is known in association with a jaw (Burrow 2004): we have rephrased this section to make it clearer

L.100 & elsewhere... lots of elsewhere... try not to overuse 'element'. If a more informative word such as 'bone' or 'cartilage' could be substituted, then do so.

We have gone through the text to address this.

L.167 – suggestion – the circular ridge (Taemasacanthus) seems far more likely to mark an attachment for a labial ligament.

This is a good suggestion, which we have incorporated into the text.

L. 186 – Fig. 1 part 'd' – colour code the teeth consistently with parts 'a' and 'b'. In the present version there appear to be only two rows of teeth: lateral and lingual. Teeth on the mesial ridge are barely visible.

We have replaced Fig 1d to keep the colour coding consistent. The teeth of the medial row are far smaller and less distinct than those of the other rows, and we have made this more explicit in the text.

Italicize taxon names in caption – here & elsewhere.

In our version of the manuscript these are italicised, so maybe this is a MS Word issue. If the paper is accepted we will keep an eye out for this in the proof.

L. 209-210. Perichondral tissue – is this bone?

We have replaced the word "tissue" with "bone".

L. 217 – what is a typical (although tipless) ischnacanthid process? And, according to whom is this process characteristic (uniquely?) of ischnacanthids? Burrow? Long?

We have rewritten this section to make this clear and have cited a relevant study.

L. 235 – Fig.2 part 'd' colour code the teeth consistently with parts 'a' and 'b' (again). Add caption detail to identify anterior (distal) and posterior (proximal) ends of bone. Add reference to the insert diagram of possible bone locations in jaws.

We have made these modifications to this and other anatomy figures.

L. 259. No medial ridge identified in the figure

We have identified this in the figure

L. 260. Only three cusps visible – and a large crack where a fourth might be?

The third cusps is truncated dorsally where it is exposed on the surface and therefore difficult to make out, and we have clarified this in the text. We have also updated the figure to colour code the cusps as for Taemasacanthus and Atopacanthus.

L. 301. In the figure, the teeth are not histologically distinct from the bone forming the

perichondral sheath of Meckel's cartilage. This might be an artefact of pdf figure quality (enlarging the image doesn't improve resolution).

Perhaps a line diagram showing the authors' interpretation of the CT histology is needed? This anatomy has been a challenge to figure using a single tomogram. It is clearer when scrolling through the CT slices, which we have provided with the paper. We have expanded the text description to make it clearer what is visible in the tomograms and what we consider to be sensible interpretations. We hope this makes the features easier to identify.

L. 306. I don't see the overlap between teeth.

Again this is difficult to figure. The overlap is extremely slight and we are tentative about its definite existence. We have changed the text to make this clearer.

L. 391. Erratic use of bold text.

The intention of these was to add mini-subheadings as we reviewed different structures per paragraph. We would prefer to keep them, as reviewer 3 flagged that they were helpful for explaining the confusing terminology associated with acanthodian dental structures.

L. 407, 433, 447, 456, 468, 486, 507 – bold text – are these subheadings? Some at start of lines; some embedded within sentences?

See above

L. 398. Possible exception? Are there reasons to challenge the teeth described in *Tetanopsyrus* (ref.24)?

We discuss this in the "occlusal plates" section later on and have amended the text to highlight this.

L. 433. Paragraph starting 'Some chondrichthyans have teeth that are not organized into files but...' – muddled structure. This section seems to be about teeth that lie directly on the jaw cartilage – file organization seems secondary. Might be worth rewriting.

We have rewritten this sentence to make this clearer.

L. 566-568. At last! Here's the pay-off, but it's buried deep in the discussion.

As recommended we have moved this into a more prominent part of the discussion as part of this reviewer's suggested rewrite. And it is also in the new title.

L. 575. Not quite sure about the meaning of 'likely dermal in origin': explain.

We have tried to rephrase this to make it clearer.

L. 596. 'borne' rather than 'bone'.

We have replaced this

L.707. Ref. 7 - add doi: 10.1038/s41559-021-01458-4

We have added this

L. 760 Ref. 23 - add the correct doi -this one links to a different ref. (Euthacanthus)

We have added the correct doi

Appendix B

Associate Editor Comments to Author:

Comments to the Author:

Thank you for engaging with the reviewer comments. As you'll see there, remain concerns from one of the reviewers regarding the work; however, given that - in the earlier round of reviews and now in this iteration - a number of reviewers were in favour of publication following revisions, we are going to ask you to conduct a further round of revision before accepting the paper (assuming the editors are satisfied by the changes made, and your rebuttals of critiques). Whether the paper is as problematic as the more critical reviewer suggests, we are of the view that it would be more productive for the paper to be available to the community, who can then engage with the work and - if it appears needed - offer a formal rebuttal: the journal encourages open debate and discussion of the published literature.

Many thanks to the editors and the reviewers for your comments and feedback. We have tried to address all reviewer comments satisfactorily in the following responses document, and have made changes in the associated manuscript. We disagree with many of Reviewer 1's comments and believe they have fundamentally misinterpreted our arguments, but we have tried to engage with their review and make changes where possible. We believe comments from all reviewers have helped to improve the manuscript and we thank them for taking the time to give detailed feedback over multiple rounds of revisions.

Reviewer comments to Author:

Reviewer: 1

Comments to the Author(s)

The most harmful problem of this manuscript is that the authors refuse to present a complete picture of what has been known about early gnathostome dentitions, but only selected information has been shown. For instance, they clearly don't want to take on board the significance of the transverse organisation (file-like arrangement) of the dentitions in stem osteichthyans and acanthothoracid stem gnathostomes, which is shared with chondrichthyan tooth whorls and tooth files, but distinct from the acanthodian dentigerous jaw bones. Consequently, the authors produce a false image that tooth whorls and dentigerous jaw bones can be united as a synapomorphy of chondrichthyans, based on the characters that are in fact general to the gnathostome total group. This will distort the understanding of dental evolution.

We disagree with the reviewer's assessment. We agree that the transversely organised files in some osteichthyans and some stem-gnathostomes are interesting and relevant, and probably represent a plesiomorphic means of tooth patterning in some sense. We refer to and cite this several times in the text both in the introduction and discussion. However, we do not agree with the reviewer's assertion that these are indistinguishable from chondrichthyan tooth whorls and files, as we discuss in the text.

We also refute the suggestion that we 'produce a false image that tooth whorls and dentigerous jaw bones can be united as a synapomorphy of chondrichthyans, based on the characters that are in fact general to the gnathostome total group'. Figure 7 summarises the distribution of oral structures including tooth whorls and dermal jaw plates, and in both cases we illustrate that these features are known outside of chondrichthyans – and indeed are absent in many chondrichthyans, including those often resolved at the base of the chondrichthyan total group. We explicitly state in the text that tooth whorls are known outside of chondrichthyans, as well as highlighting the resemblance between dentigerous jaw bones and

the jaw bones of stem-gnathostomes (although we do not explicitly draw homology between these structures).

Although this manuscript focuses on acanthodians, a much more serious (though can be brief) review of various early gnathostome dentitions is expected in the Introduction/Discussion. The authors should not just cite a bunch of papers and then run away, saying that other taxa or oral structures are poorly understood, poorly characterized, etc.

We do not believe we “cite a bunch of papers and run away”. We give a brief overview of morphologies in non-chondrichthyan taxa, giving the reader key citations to follow. We consider this sufficient for an MS focussed firmly on chondrichthyans, a change that was made following requests by other reviewers (particularly reviewer 4) in the previous round of revisions.

Line 21

“parallel, continuously replacing” is not very informative. It means nothing for people that are not familiar with shark dentitions. I suggest you clarify the transverse organisation of the files and the conveyer-belt replacement.

We have amended the text to clarify this.

Line 22

the site-specific replacement?

We have amended the text to clarify this.

Line 54

What do you mean by “partially” here?

We have amended the text to clarify that we mean that the tooth base is partially resorbed.

Line 62-64

The dentitions of stem gnathostomes are definitely relevant to the evolution of teeth, no matter how “strange” they are. They are not poorly characterized, and we do have understood quite a bit.

We do not state that these taxa are not relevant to the evolution of teeth, nor that those that have been described are poorly characterised. Rather, we are saying that the oral structures in many Palaeozoic taxa – including those that are the focus of this paper – are poorly characterised, and as a result their relevance to the origin of teeth is poorly understood. We have revised this sentence to make our meaning clearer.

Line 77

This is not an accurate description of the early osteichthyan dentitions.

We are unsure in which sense the reviewer considers this to be an inaccurate description of Palaeozoic osteichthyan dentitions.

Line 90-91

The long history of file-like arrangement can be traced back to the basal jawed stem gnathostomes. The file-like arrangement also occurs in stem osteichthyans, rather than limited to chondrichthyans.

We state this in the summary of dentitions in the previous paragraph.

Line 94-95

at the labial/outer jaw margin

We have added the word 'labial' to make this clearer.

Line 98-101

“file-like teeth” and “file-like whorls” sound strange. Does the former mean files of teeth that are not fused into a whorl (non-joined teeth), or does it include the whorl condition? “File-like teeth” sounds like a description of tooth shape. Whereas all tooth whorls bear tooth files, in other words, there are no “non-file-like whorls”. Also, there is an intermediate condition that the teeth are not fused by a common base, but post-functional teeth are retained and packed at the outer margin of the mouth, which looks like a tooth whorl. Shedding or non-shedding is a key difference of the different conditions of file-like organization, and thus should be mentioned in the description of the different conditions. Yes, it has been mentioned in the following text, but it just makes it more confusing for readers who are not familiar with early chondrichthyan dentitions. So far, not a simple definition of tooth whorl has been given, but readers should not be supposed to know what is a tooth whorl.

We have removed the initial reference to a tooth whorl to avoid confusing the reader, and have revised the sentence to make it clear that we are referring to teeth arranged into files rather than a tooth shaped like a file. We have also added an explicit definition of a tooth whorl. The intermediate condition is referred to in the following sentence.

Line 106

Not sure which mode of tooth replacement you are talking about. Not understand why the topic of file-like organization suddenly becomes the mode of tooth replacement, and then jumps to tooth whorls in the next sentence. As no osteichthyan is mentioned in this paragraph, there should not be a second mode of tooth replacement, but conveyor-belt style tooth replacement.

We have revised this and removed mention of the whorls early on in the paragraph to avoid confusion.

Line 120-121

It is also unclear whether tooth whorls are shed from time to time.

We have incorporated this into the text.

Line 132-135

This mode of growth has long been well understood since Ørvig 1973, which is based on several identified taxa. Then Smith 2003. This statement is acceptable for Rücklin et al. 2021 only if say “have been shown by microtomography”. Since this is a common mode of tooth addition in acanthodian dentigerous jaw bones, it is not that important whether the specimens are identified or not.

The reviewer is right that this mode of growth was proposed by Ørvig and later Smith, but it wasn't well-characterised in 3D until the work of Rücklin et al, who were able to confirm that this mode of growth occurred over the competing hypothesis that the bone was shed *in toto* using synchrotron tomography. We have added citations for Ørvig and Smith to this section. However, we maintain that other aspects of the construction and morphology are poorly characterised.

Regarding the second point, we disagree that it is not important whether specimens are identified or not. It is important that observations be made in taxa that also preserve

articulated remains in order to link isolated material to states in completely-known taxa and incorporate these into phylogenetic analysis. Furthermore, growth may be different even in closely related taxa. Making assumptions about the gross morphology of taxa which are only known from fragmentary remains is dangerous given the numerous unknowns of the early vertebrate fossil record.

Rephrase “wider construction”.

We have rephrased this to make it clearer what we mean

Line 137-138

Strange sentence. Are you trying to say “teeth that are not borne by dentigerous jaw bones”? “not part of ... tooth files” is redundant, when saying “not arranged into files”.

This sentence was poorly written: we have rephrased.

Line 228

“gnathal plates” is usually used in placoderms. It is better to stick to “dentigerous jaw bones” for ischnacanthids, or just say “jaw bones” for acanthodians regardless it is dentigerous or not.

We agree with the reviewer and have changed the text in this case.

Line 264-265

"Elgin" is not a known Middle Devonian locality. In fact, there are no Middle Devonian localities at all around Elgin, only Late Devonian ones such as Scaat Craig and Rosebrae. I wonder where this specimen is from.

This is a typo on our part and we have changed the text. Thanks for catching this.

Line 368-371

Citation of figures is needed. Why does its position suggest an artefact of growth?

We have removed this sentence from the text

Line 381-383

Shown in any figures?

We have labelled the medial tooth row in the supplementary figure, which shows the vasculature.

Line 436-437

Why is it flatter but taller?

We meant flatter laterally and taller but this was not clear. We have amended the text.

Line 463

Is the inner tooth row the same as the lingual tooth row? If so, keep the consistency of the terms. Otherwise, do you mean the inner one of the two rows borne by the medial ridge?

Yes it is – we have rewritten to keep consistency of terms.

Line 465-468

Label the two additional small cusps in the figures. Do you mean they form an additional tooth row?

We have labelled them in the figure and included a callout in the text. We do not think it is appropriate to describe them as a row as there are only two of them.

Line 550-552

Bad sentence construction.

We have rewritten this sentence.

Line 641-642

Do you mean the radial vasculature? Is the point, which the vasculature outwards from, the ossification centre of both the dermal and endoskeletal bones? Make it clearer with shorter sentences.

We have split this sentence and rewritten it to make it clearer that we mean the radial vasculature and refer to growth in the dentigerous jaw bone. We can make no inferences about the ossification centre of the endoskeletal bone, as it is not sufficiently well-preserved.

Line 662-675

The chondrichthyan dentitions with tooth whorls lining along the length of jaw and whorl-like cheek scales are not site-specific either, as in acanthothoracid stem-gnathostomes and early osteichthyans. All those suggest common patterning mechanisms across early gnathostomes, rather than an apomorphic for chondrichthyans. In other words, many other dental elements across gnathostomes can be regarded as “stretched out” tooth whorls, with new teeth added in various orientations, such as lungfish tooth plates.

It is necessary to define “site-specific” here. Is the site relative to the whole jaw or mouth, or to a dermal plate or an element that unite the teeth? It is especially confusing because you also use “site-specific” to describe the dentitions of bony fishes in the abstract.

We don't disagree that “adding cusps in a row” is common patterning mechanism across gnathostomes, e.g. in this paragraph “*Teeth and tooth-like structures are added directionally across the gnathostome total-group and this may be a plesiomorphic feature of gnathostome dentitions*”. However as we explain in the text we consider that chondrichthyan tooth files are united by an organisation onto specific locations on the underlying dermal bone, rather than into haphazardly arranged files of teeth in acanthothoracoids and early osteichthyan marginal jaw bones, even if they share an underlying mechanism of directional addition of new cusps. Similar, organised files are indeed seen in early osteichthyan tooth whorls and e.g. in lungfishes. Lungfishes are sufficiently phylogenetically remote from chondrichthyans, and with well-known relatives without toothplates, for their mode of growth to be identified as homoplasious. Palaeozoic osteichthyan tooth whorls are an interesting comparison, but have been shown to have a distinctly osteichthyan mode of growth (Doeland et al 2019) and are unlikely to be homologous with those of chondrichthyans (Rücklin et al 2021).

However the reviewer makes a fair point that using the terminology “site-specific” is confusing as it implies that they resemble the osteichthyan condition. We have rewritten this section to resolve that, and more clearly state our interpretation.

You are trying to find a dental character to unite the chondrichthyans. It is nice if there are any. But if not, don't make it up. It is not helpful to understand early gnathostome evolution.

We refute the suggestion that we are making things up.

As you said “Support values aside from the chondrichthyan total group and crown-group nodes are typically low”, in addition to the situation that the osteichthyan-chondrichthyan node is difficult to place among stem-gnathostomes, it is unwise to put things into boxes and seal the boxes. Early vertebrate phylogeny should be based on morphological observations, instead, your interpretation of the morphology is determined by the phylogeny, which is very unstable. Although all acanthodians are considered stem chondrichthyans at the moment, the diversity of early chondrichthyan dentitions is supposed to cast light on the potential evolutionary relationship between acanthodians and other early gnathostome groups. If acanthodians, at least some of them, were placed in the osteichthyan stem, as in the old times, do you still think that the dentigerous jaw bones are just a modified form of tooth whorls, or more similar to tooth whorls than any other kinds of gnathostome dentitions? In terms of “site-specific”, the linear tooth rows of osteichthyans are more comparable, such as the row of fang pairs, the row of marginal large teeth and the row of accessory small teeth are all site-specifically arranged.

Our observation of the morphology itself is not determined by phylogeny, but it does inform how we interpret the evolution of that morphology. We don't characterise dentigerous jaw bones as “just a modified form of tooth whorl”, instead we offer an interpretation that might help us understand their morphology in the light of what we see in other chondrichthyans, given the evidence that these bones belong to stem-chondrichthyans. We are also cautious to offer alternative hypotheses, given the instability of relationships.

We are unsure if the reviewer is actually suggesting that these things are stem-osteichthyans, but as it stands the weight of evidence is all ‘acanthodians’ are chondrichthyans (a very well-supported node including in the analyses in this paper) and this paper is based on that premise. So we consider it reasonable on our part not to discuss the possibility that they are osteichthyans. Should this latter hypothesis re-emerge in subsequent analyses, we would then reinterpret the evolution of dental morphologies accordingly.

Developmentally, it does not make sense for the dentigerous jaw bones, a kind of “antero-posteriorly ‘stretched out’ tooth whorls” in your words, to evolve from the labio-lingual tooth whorls.

We don't argue that one evolved directly from the other, we argue that they share a common patterning mechanism. And teeth obviously can undergo pretty major transformations in organisation during their evolution, for example holocephalan or lungfish tooth plates.

Line 735-736

The tooth whorls of iniopterygians is more likely a retained primitive character, because many other stem holocephalans also have tooth whorls along the jaw.

Iniopterygians and eugeneodontids are stem-holocephalans which have tooth whorls: both of these taxa have very strange anatomies and the whorls are restricted to the symphyseal area of the jaw. Given that there are generative series of teeth with separate bases symmoriiform and cladoselachian sharks (stem-holocephalans less closely related to the crown-group (e.g. Coates et al. 2017)), and in stem-holocephalans probably more closely related to the crown-group like *Debeerius*, *Helodus*, and petalodonts we consider it more likely that the tooth whorls of iniopterygians and eugeneodontids are not a retained primitive character.

Line 872-879

Do you mean the dentition entirely formed from tooth whorls share a last common ancestor

and develop independently in the crownward lineage is one of the scenarios? It is confusing what scenario Frey et al. suggested, according to your statement.

We have amended this sentence to try and make this clearer.

As one of the major types of dentitions in Acanthodii, why do you say it “being lost in Acanthodii”?

We meant Acanthodii *sensu* Coates et al. 2018, i.e. the possible clade of things including diplacanthiforms, ischnacanthiforms, and acanthodiforms, none of which have a dentition entirely formed from tooth whorls. We have amended the text to clarify this.

Line 896-899

It will be nice to indicate the clade only with tooth whorls and the clade with a diverse array of oral structures at the nodes of the trees if they do form different clades.

We have avoided labelling nodes with respect to individual characters in order to avoid overly confusing the figure. States in different taxa in the phylogeny are shown in the table on the right hand side.

Figure 1

In (a), it looks like each tooth is a “tooth file”. It is better to box the file of teeth and label the box with a single line. It will be nice to have an inset of the lateral view just like in (b) for comparison.

Abbreviations should be explained in the caption. But it is better not to use abbreviations at all since there is enough space, and the consistency should be kept (“tooth file”, “tooth whorl” etc. are not abbreviated).

We have added a close up tooth series (and changed terminology to better match new text) to the figure as suggested. We have also amended the labels to avoid abbreviations as recommended.

Supplementary figure 1

Which taxon does (d) belong to, Taemasacanthus or Atopacanthus? No need to correspond the taxa with the figures twice in the caption, which actually brings up a conflict. (a) and (d) can be both medial views if they belong to the same taxon. (b) is from the box of (a), rather than the lateral view, so all the following figures are wrong. Same for (f). Check the figure citations in the main text. It is also unclear which figures share the same scale bar. Since all taxa are given the same colour, it is better to divide the figures into sections for each taxon by adding some separating lines and labelling each division with the taxon names.

The caption and figure have been adjusted accordingly and citations of this figure have been checked in the main text.

Can the circular ridge be seen in any of the figures? If so, please label it.

We have added a panel to figure 2 and referred to this in the text.

Reviewer: 4

Comments to the Author(s)

Review 2– (new title)

Diverse stem-chondrichthyan oral structures and evidence for an independently acquired acanthodid dentition.

Dearden, R. P. & Giles, S.

General comments.

Much improved. The included comments are mostly minor and easily corrected. A useful and now, content-wise, distinct contribution to the study of shark origins, early evolution of jawed vertebrates, and the phylogeny of dental systems.

Detailed comments.

Introduction

L58-61. Clarity: are the dentitions or bones arranged radially?

We have rewritten this part to be clearer

The dermal bones of what taxon are uncertainly homologous to crown gnathostome dental arcades?

We have rewritten this part to be clearer

L61-66. Another sentence that needs sorting out. Something about the early history of early osteichthyans – too many ‘early’s creates ambiguity – is this about ontogeny?

We have changed the text to state “Palaeozoic” a rather than “early to avoid this problem.

L85. Teeth not ‘tooth’.

We have rephrased this sentence in line with reviewer 1’s comments.

L87. indicates not ‘indicating’

We have rephrased this sentence in line with reviewer 1’s comments.

L88. taxa or clades, not ‘animals’

We have changed this to taxa

L103. Other not ‘others’

We have made this change

L104. Italicize ‘Gladbachus’.

We have made this change

L271 and elsewhere: check figure numbers in case there’s a general slippage. Here, ‘1f’ should be ‘2f’.

We have checked figure numbers for general slippage and have corrected any mistakes.

L272. Remove ‘ref’.

We have made this change

L288. 1b should be 2b.

We have made this change

L386. ‘beunmineralized’

We have made this change

L453. Italicize ‘Doliodus’

We have made this change

L458. Decay is a process. The Bayesian analysis is not a process or treatment of the max. parsimony results.

The Bayesian analysis simply failed to find the Acanthodii clade in some or any trees, sufficient to register in the maj. rule summary.

We amended the text to change this

L468. Orvig didn’t ‘suppose’ – he hypothesized, estimated, or conjectured.

True: we have rewritten this so that he hypothesized instead.

L485. I like the notion that toothwhorls are dentigerous jaw bones rolled-up. Very satisfying.

L518. Triangular or conical?

We have changed this to read ‘conical’

L519. Cryptic homology? Oh dear. Sort this out – what do you mean? Something is homoplastic, but what? And is some other part of this system thought to be homologous?

We have removed the offending text and rewritten this sentence to clarify.

L531-532. (except for Acanthodopsis) – no parenthesis required.

We have made this change

L545-549 – neither of the refs cited is original on this matter (#65 is third-order review/summary). Suggest a ref. from Moya Smith, perhaps?

We have replaced this reference as recommended (Smith and Coates 1998)

L552 – more conventionally shark-like taxa?

We have made this change

L572. Un-bold the text.

We have tried to use bold text in these paragraphs to highlight what each one is about, so we would prefer to leave it in. A previous reviewer also highlighted that this helped them as a non-specialist. If that isn’t in keeping with the editorial policies of the journal though we are happy to unbold it.

L576. present, not presents

We have made this change

L582. That that

We have made this change

L587-592. First to third sentences – there’s a simple systematic statement buried within but obscured by ...words. Re-think; re-phrase.

We have rephrased this part of the paragraph.

L613. Comparative to what? This piece of string is relatively long.

We have removed this part of the sentence as it wasn't necessary information for the paragraph and would have required a lot more text to state clearly.

L635. More animals: remove.

We have replaced this with "taxa"

L.688+ Tooth shedding scenario – yes and no. Don't forget the presence of generative tooth series in stem chimaeroids such as Debeerius and Helodus. Are these tooth shedders or retainers?

We have rewritten the text in this to accommodate this useful point.

Extra typos – initiall & chondrichthyan

We have corrected these typos

L698. Shedding evolved multiple times. Yes. Good point & worth repeating.

L705. Frey et al include a more limited suite of stem chondrichthyans, thus deficient?

We have now mentioned this in the mention of Frey et al. in the previous paragraph.

L713. A period of inferred or estimated rapid gnathostome evolution.

We have incorporated 'inferred'

7L14. 'in of'

We have corrected this

L720+. Section 4. Conclusion. Cut? Redundant text unless a summary is required, in which case relabel as 'Summary'.

We have relabelled this 'summary'